# Single-cell transcriptomics identifies CD44 as a marker and regulator of endothelial to haematopoietic transition

Morgan Oatley[1,12], Özge Vargel Bölükbası [1,7,12], Valentine Svensson [2,3,8], Maya Shvartsman[1], Kerstin Ganter[1], Katharina Zirngibl[4], Polina V. Pavlovich[1,5,9], Vladislava Milchevskaya[4,10], Vladimira Foteva[1], Kedar N. Natarajan [2,11], Bianka Baying[6], Vladimir Benes [6], Kiran R. Patil [4], Sarah A. Teichmann [2] & Christophe Lancrin [1]*

The endothelial to haematopoietic transition (EHT) is the process whereby haemogenic endothelium differentiates into haematopoietic stem and progenitor cells (HSPCs). The intermediary steps of this process are unclear, in particular the identity of endothelial cells that give rise to HSPCs is unknown. Using single-cell transcriptome analysis and antibody screening, we identify CD44 as a marker of EHT enabling us to isolate robustly the different stages of EHT in the aorta-gonad-mesonephros (AGM) region. This allows us to provide a detailed phenotypical and transcriptional profile of CD44-positive arterial endothelial cells from which HSPCs emerge. They are characterized with high expression of genes related to Notch signalling, TGFbeta/BMP antagonists, a downregulation of genes related to glycolysis and the TCA cycle, and a lower rate of cell cycle. Moreover, we demonstrate that by inhibiting the interaction between CD44 and its ligand hyaluronan, we can block EHT, identifying an additional regulator of HSPC development.

[1] European Molecular Biology Laboratory, EMBL Rome - Epigenetics and Neurobiology Unit, via E. Ramarini 32, 00015 Monterotondo, Italy. [2] Wellcome Trust Sanger Institute, Wellcome Genome Campus, Hinxton, UK. [3] European Molecular Biology Laboratory, EMBL-EBI, Wellcome Genome Campus, Hinxton, Cambridgeshire CB10 1SD, UK. [4] European Molecular Biology Laboratory, Structural and Computational Biology Unit, Meyerhofstrasse 1, 69117 Heidelberg, Germany. [5] Moscow Institute of Physics and Technology, Institutskii Per. 9, Moscow Region, Dolgoprudny 141700, Russia. [6] European Molecular Biology Laboratory, Genomics Core Facility, Meyerhofstrasse 1, 69117 Heidelberg, Germany. [7] Present address: Stem Cell and Regenerative Biology Department, Harvard University, 7 Divinity Avenue, Cambridge, MA 02138, USA. [8] Present address: Pachter Lab, Division of Biology and Biological Engineering, California Institute of Technology, 1200 East California Boulevard, Pasadena, CA, USA. [9] Present address: Max Planck Institute of Immunobiology and Epigenetics, Stübeweg 51, D-79108 Freiburg, Germany. [10] Present address: Institut für Medizinische Statistik und Bioinformatik, Bachemer Strasse 86, 50931 Köln, Germany. [11] Present address: Department of Biochemistry and Molecular Biology, The University of Southern Denmark, Danish Institute for Advanced Study, Campusvej 55, 5230 Odense M, Denmark. [12] These authors contributed equally: Morgan Oatley, Özge Vargel Bölükbası. *email: christophe.lancrin@embl.it

Understanding the developmental origin of haematopoietic stem and progenitor cells (HSPCs) is of critical importance to efforts to produce blood and blood products in vitro for medical applications. HSPCs originate from endothelial cells in the aorta-gonad-mesonephros (AGMs) of mid-gestation embryos[1,2]. This process known as the endothelial to haematopoietic transition (EHT) requires drastic morphological changes that have been directly visualized through time-lapse imaging studies both in vitro and in vivo[3–6]. EHT is a highly conserved process that has been studied across vertebrate models from *Xenopus* and zebrafish to mice[7]. Importantly, the human definitive blood system also has an endothelial origin[8].

The best tools so far to detect endothelial cells with haemogenic capabilities rely on using fluorescent reporters under the control of *Runx1*[9] or *Gfi1*[10] regulatory elements, two key transcription factors in the process of EHT. Cells expressing these transcription factors already co-express blood and endothelial genes. However, we still do not know the nature of the endothelial cells that will acquire the expression of these transcription factors. Nor do we know whether any endothelial cell in the AGM can initiate the haematopoietic programme or if a certain type of endothelial cell is primed to undergo EHT.

Despite our lack of characterization of the definitive precursor to HSPC development, haemogenic endothelium (HE), recent advances have been made in terms of in vitro HSPC generation via an endothelial intermediate. Through the use of transcription factor cocktails, both human pluripotent stem cells via a HE stage and adult mouse endothelial cells have been successfully reprogrammed into multi-potent, definitive haematopoietic stem cells (HSCs)[11,12]. However, the use of endothelial populations in the process emphasizes the importance of an improved understanding of HE.

The early haematopoietic hierarchy has been described as a three-step process based on phenotypic characteristics. Specifically, Pro-HSC, Pre-HSC type I and type II populations have been defined based on their expression of the cell surface markers CD41, CD43 and CD45, as well as the time taken to mature into definitive HSCs in OP9 co-culture[13]. Recently, an in-depth transcriptional investigation was performed on the type I and type II Pre-HSC populations at day 11 of mouse embryonic development; however, the earlier stages of EHT remain largely uncharacterized[14]. Indeed, gaining a solid understanding of HE and the initial steps that endothelial cells must take to become HSPCs has proved difficult in the absence of a robust marker.

Through antibody screening and single-cell RNA sequencing (sc-RNA-seq), we identified CD44 as a marker of EHT, enabling the isolation of key cellular stages of blood cell formation in the embryonic vasculature. CD44 is a cell surface receptor principally involved in the binding of the extracellular matrix molecule hyaluronan[15]. Its cell surface expression has been used to identify cancer stem cell populations and has been strongly linked to the metastatic potential of many cancers[16–20]. Although previous research has revealed the importance of CD44 in HSPC migration to the bone marrow, the role of the receptor in early embryonic haematopoiesis has not been characterized[21]. Using CD44 expression in conjunction with VE-Cadherin (VE-Cad) and Kit, we are able to clearly differentiate between the different types of AGM endothelial cells, Pre-HSPC-I and Pre-HSPC-II more accurately than using the combination of VE-Cad, CD41, CD43 and CD45 markers. This has allowed us to perform extensive transcriptional profiling, making it possible to characterize the very earliest changes in haematopoietic differentiation from endothelial cells. Moreover, by disrupting the interaction of CD44 and its ligand, we could inhibit EHT, demonstrating an unexpected role for CD44 in the emergence of HSPCs.

## Results

**CD44 is a potential marker of early haematopoietic fate**. HSPCs have an endothelial origin within the embryo[4–6,22]. To better characterize the EHT, we performed both an in vitro antibody screen and an in vivo sc-RNA-seq experiment to identify markers allowing the identification of subpopulations within VE-Cad+ endothelial cells (Supplementary Fig. 1). Antibodies against 176 cell surface markers[23] were tested against the VE-Cad+ population generated from the in vitro ESC differentiation system into blood cells[24]. Forty-two of these markers were expressed on VE-Cad+ cells (Supplementary Table 1). We looked for bimodal expression to separate distinct endothelial populations and identified a shortlist of 16 candidates to test in vivo, including CD41 and Kit, already known to split the VE-Cad+ cells[13]. Six of these markers (CD44, CD51, CD61, CD93, MadCam1 and Sca1) were found to split the VE-Cad+ endothelial population of the AGM in two (Fig. 1a and Supplementary Fig. 2). In parallel, we used sc-RNA-seq to analyse the transcriptional profiles of VE-Cad+ cells from the AGM region at E10.5 (Fig. 1). Clustering analysis identified a population with both haematopoietic and endothelial gene expression, distinct from the other endothelial population (Fig. 1b and Supplementary Data 1). Bioinformatics analysis showed that *Cd44* is one of the best marker genes for this population of transitioning cells co-expressing endothelial and haematopoietic genes (Fig. 1c). The expression of *Cd44* was also positively correlated with other known haematopoietic markers such as *Runx1*, *Gfi1* and *Adgrg1* (*Gpr56*) (Fig. 1d). Given the association of *Cd44* with endothelial cells undergoing EHT at both the protein and mRNA level, we decided to further investigate its role in embryonic haematopoiesis.

**CD44 marks different cell populations in the AGM**. To validate our screening results and investigate the identity of CD44+ cells, we performed immunofluorescence and more detailed flow cytometry analysis on the AGM region of mouse embryos (Fig. 2). Immunofluorescence of cross-sections of mouse AGMs revealed that CD44 marked cells that were part of the vascular wall and cells that were incorporated in haematopoietic clusters at E10 and E11 (Fig. 2a and Supplementary Fig. 3). Different levels of CD44 expression could be noticed including some parts of the arterial wall being negative for this marker (Supplementary Fig. 3). Flow cytometry revealed that CD44 expression significantly increased in the VE-cad+ endothelium of the AGM between E9.5 and E10.5 when cells are undergoing EHT (Fig. 2b, c). Furthermore, by staining with an antibody against Kit (a marker of intra-aortic haematopoietic clusters)[25], we found that a large proportion of cells with lower levels of CD44 expressed little or no Kit (Fig. 2d).

By grouping the cells based on their expression of CD44 and Kit, we found these populations to be significantly different in terms of cell size, suggestive of cells undergoing a morphological transition (Fig. 2e). Altogether, our results indicate that CD44 marks a subset of endothelial cells and cells in the haematopoietic clusters of the AGM during the key window of HSPCs development in the mouse embryo.

**Detection of two CD44+ populations expressing blood genes**. Using the Biomark HD single-cell quantitative PCR (qPCR) platform, we analysed the expression of 95 genes associated with both endothelial and haematopoietic cell types[24]. We performed extensive transcriptional profiling on the CD44Neg, CD44Low-KitNeg, CD44LowKitPos and CD44High populations identified (Fig. 2d) between E9.5 and E11.5 (342 cells in total). We found that the CD44LowKitPos and CD44High populations expressed

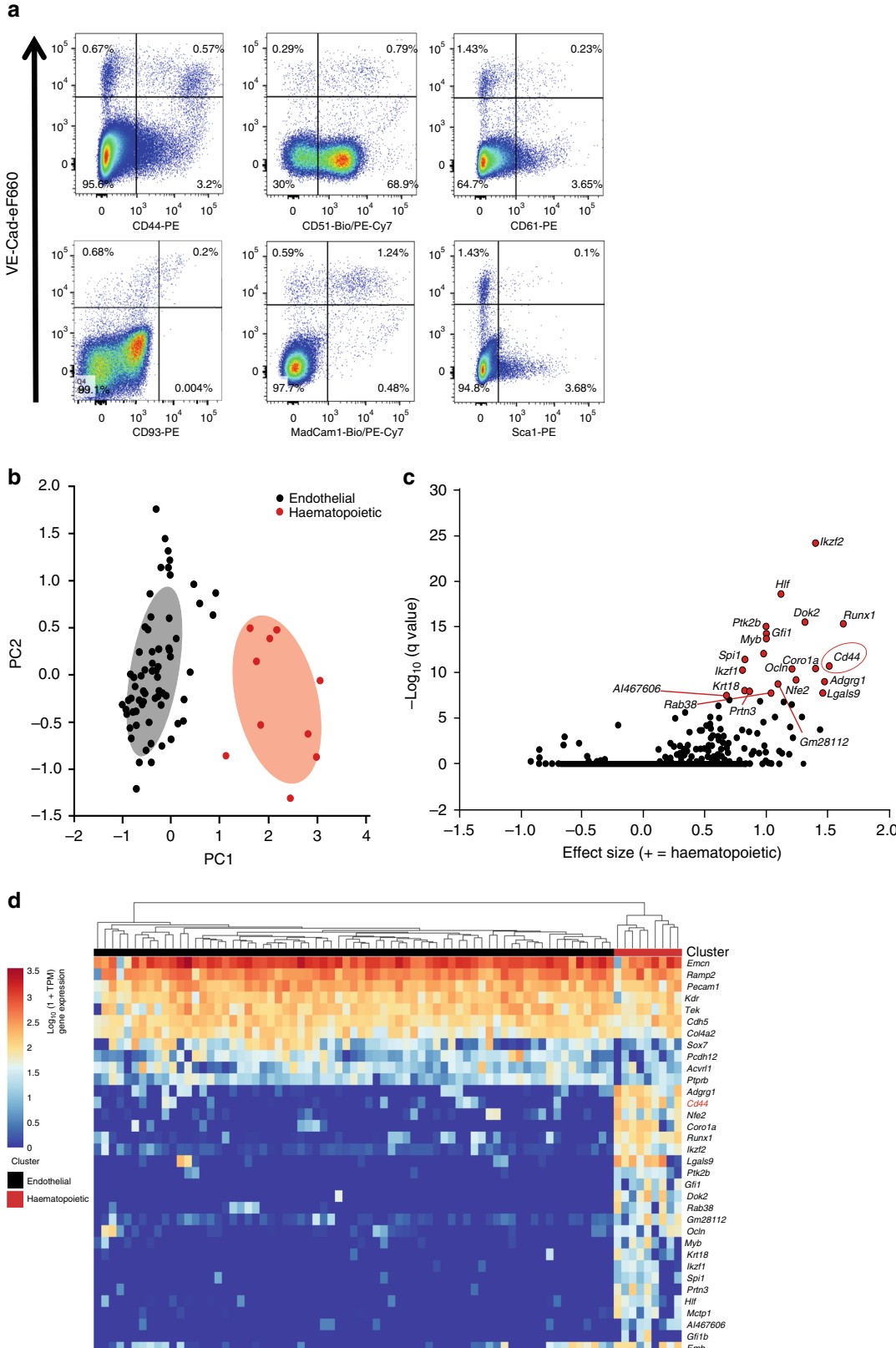

**Fig. 1 Search for markers to dissect the endothelial to hematopoietic transition. a** FACS plots of cells isolated from the AGM region at E11, stained with VE-Cad and indicated cell surface markers selected from the antibody screen. **b** Principal component analysis of the single-cell RNA-seq data done at E10.5. Cells expressing haematopoietic genes are marked in red, while the other cells are marked in green. **c** Volcano plot showing a selection of marker genes specific to the group of cells expressing haematopoietic genes. *Cd44* is highlighted with a red circle. **d** Heatmap displaying the expression of a selection of genes in the endothelial and haematopoietic clusters. *Cd44* is highlighted in red. See also Supplementary Fig. 1 and Supplementary Data 1.

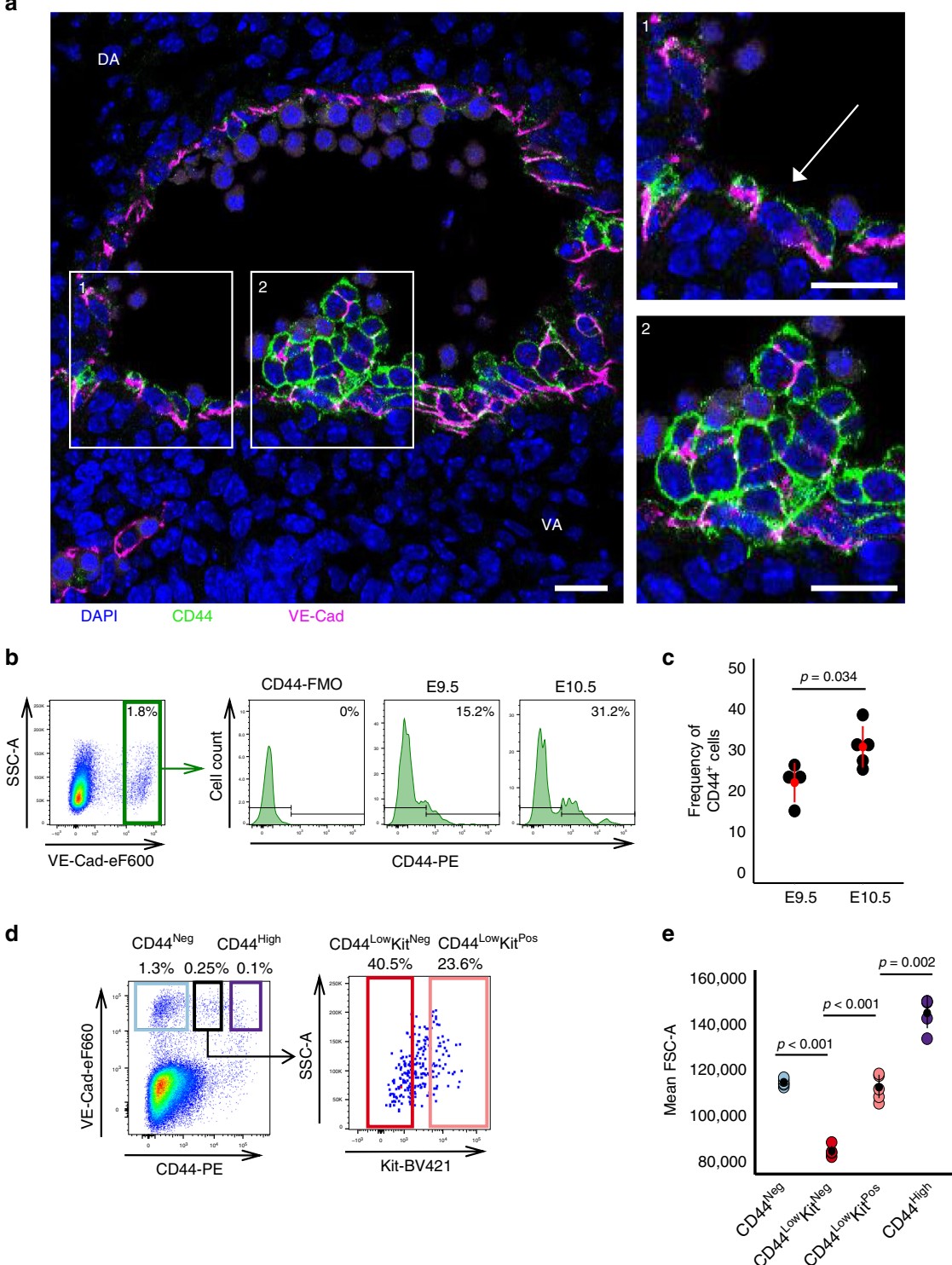

numerous haematopoietic genes (Supplementary Fig. 4 and Supplementary Data 2). The CD44^LowKit^Pos cells also expressed many endothelial genes as in our sc-RNA-seq analysis (Fig. 1d); however, the CD44^High cells appeared to be more advanced in the EHT process and lacked endothelial gene expression (Supplementary Fig. 4). Of note, the CD44^LowKit^Pos population expressed specifically *Gfi1* and *Itgb3*, as well as *Gata2*, *Runx1*, *Lyl1*, *Erg*, *Fli1*, *Lmo2* and *Tal1*, whose simultaneous co-expression is responsible for the dual endothelial-haematopoietic identity of pre-HSPCs (Supplementary Fig. 4)[24].

Conversely, both the CD44^Neg and CD44^LowKit^Neg populations showed specific endothelial gene signatures and lacked haematopoietic gene expression (Supplementary Fig. 4). Despite their different CD44 expression patterns, these cell populations clustered together (Supplementary Fig. 4). We repeated this experiment using another selection of 96 genes based on our sc-RNA-seq experiment (Supplementary Table 2 and Supplementary Fig. 5). With this gene list, we were able to confirm the dual endothelial-haematopoietic and haematopoietic identities of the CD44^LowKit^Pos and CD44^High populations, respectively.

**Fig. 2 CD44 splits the VE-Cadherin+ cells of the AGM into different populations. a** Immunofluorescence of VE-Cad (magenta) and CD44 (green) expression in a cross-section of the AGM region of a wild-type embryo at E10 (32 somite pairs). Images 1 and 2 show higher magnification of the areas highlighted in the main image, showing CD44 marking endothelial cells in the vascular wall and a haematopoietic cluster. Scale bars represents 25 μM. **b** FACS plots indicating percentage of cells expressing high levels of VE-Cad from dissected AGMs of wild-type embryos. The histograms indicate the percentage of VE-Cad[High] cells positive for CD44 at both E9.5 (28 somite pairs) and E10.5 (35 somite pairs) compared with the FMO. **c** Percentage of CD44+ cells within the VE-Cad[High] fraction, each data point represents an independent experiment and independent pooled litter of embryos, E9.5 $n = 4$ and E10.5 $n = 5$. We compared the average proportion of CD44+ cells at E9.5 ($M = 22.13$, SD $= 4.85$) to E10.5 ($M = 30.88$, SD $= 5.08$). Significance was determined by a Welch's two-sample t-test, $t(6.7) = 2.635$, p-value $= 0.0345$. **d** Representative FACS plots of gating strategy and expression of VE-Cad and CD44 in the AGM region of wild-type embryos at E10 (30–34 somite pairs). Expression of Kit cell surface marker is highlighted for the CD44[Low] population. Full gating strategy is shown in Supplementary Fig. 5a. **e** Mean FSC-A as an indication of cell size is plotted for each population (CD44[Neg], CD44[Low]Kit[Neg], CD44[Low]Kit[Pos] and CD44[High]) for five independent litters of E10 wild-type embryos ($n = 5$). Significance was determined by a one-way ANOVA, $F(3, 16) = 142.01$, p-value $< 0.0001$ followed by Tukey's HSD post-hoc tests to determine where the significance lies *$p < 0.05$, **$p < 0.01$, ***$p < 0.001$. Error bars represent SD. Source data are provided as a Source Data file. See also Supplementary Fig. 2.

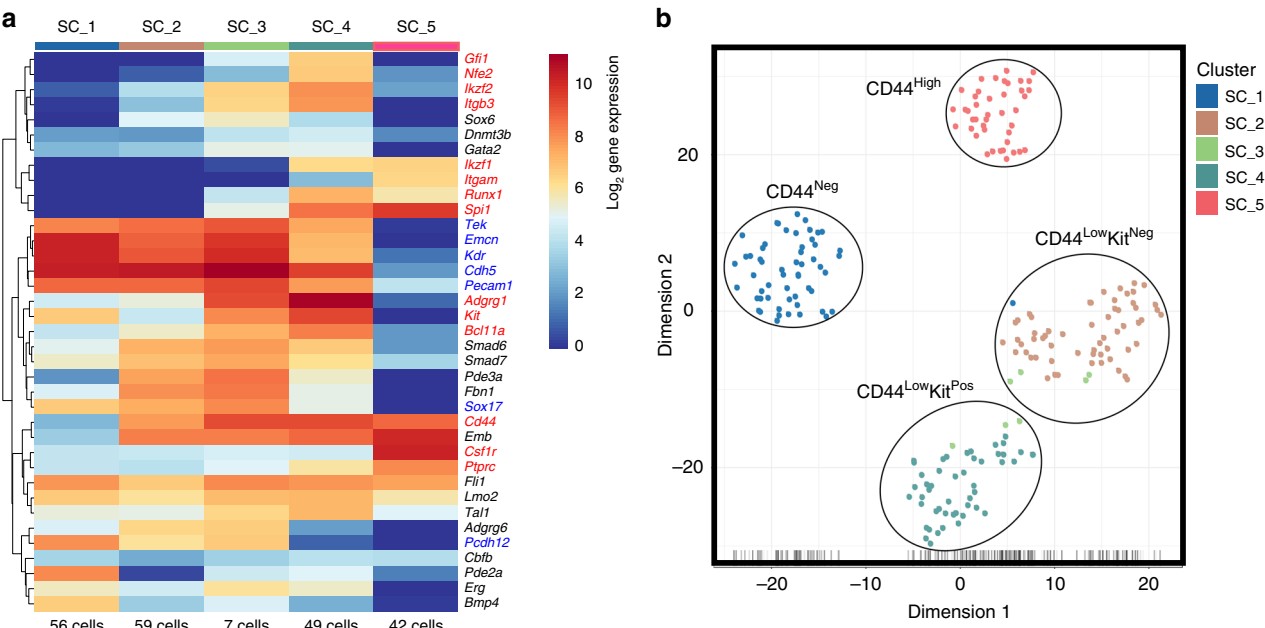

**Fig. 3 The VE-Cad+ subpopulations defined by CD44 are transcriptionally distinct. a** Heatmap showing average expression of endothelial (blue), haematopoietic (red) and various (black) genes in the indicated groups (genes are clustered using Pearson's correlation). The number of cells for each cluster is indicated at the bottom of the panel. Five groups are indicated. **b** tSNE plot from single-cell q-RT-PCR data shown in **a**. See also Supplementary Fig. 3, Supplementary Fig. 4 and Supplementary Data 3.

Surprisingly, the CD44[Neg] and CD44[Low]Kit[Neg] populations formed two distinct clusters (Supplementary Fig. 5b, Fig. 3a, b and Supplementary Data 3). In addition to *Cd44*, we found the genes *Adgrg6*, *Fbn1*, *Pde3a*, *Smad6*, *Smad7* and *Sox6* to be upregulated in the CD44[Low]Kit[Neg] cells compared with CD44[Neg]. In contrast, *Bmp4*, *Kit*, *Hmmr* and *Pde2a* were more expressed in CD44[Neg] endothelial cells (Supplementary Fig. 5b).

Although the four groups defined by VE-Cad, CD44 and Kit were confirmed to be distinct transcriptionally; our clustering analysis found a fifth population (SC_3) composed of cells from both CD44[Low]Kit[Neg] and CD44[Low]Kit[Pos] groups (Supplementary Fig. 5b and Fig. 3a). Its transcriptional profile was found to be intermediary between SC_2 (CD44[Low]Kit[Neg]) and SC_4 (CD44[Low]Kit[Pos]), e.g., it expressed *Adgrg1*, *Runx1*, *Itgb3* and *Spi1* such as SC_4 but still expressed *Adgrg6* and *Pcdh12* such as SC_2 (Supplementary Fig. 5b). Interestingly, it is within this transitional SC_3 population that we saw the upregulation of *Runx1*, *Spi1* and *Gfi1*, which are three of the four transcription factors used to reprogramme adult mouse endothelial cells into HSCs[12].

In conclusion, single-cell quantitative reverse transcriptase PCR (q-RT-PCR) analysis identified three homogeneous VE-Cad[Pos] CD44[Pos] populations (CD44[Low]Kit[Neg], CD44[Low]Kit[Pos] and CD44[High]) with an increasing haematopoietic gene expression and decreasing endothelial gene transcription. This would suggest a developmental link between the CD44[Low] populations where CD44[Low]Kit[Neg] cells would be the direct precursors of the CD44[Low]Kit[Pos] population, which would then go on to generate CD44[High] cells.

**Pro-HSC, Pre-HSC-I and Pre-HSC-II populations express CD44.** To place our results in the context of known AGM endothelial subpopulations, we performed transcriptional analysis of the Pro-HSC, Pre-HSC type I and Pre-HSC type II populations defined by the combination of VE-cad, CD41, CD43 and CD45 markers (Supplementary Fig. 6a and Fig. 4a)[13]. Gating used in Fig. 4a was determined based on fluorescence minus one controls and reflects the conditions we used to sort these rare haematopoietic populations. Although the gates were not perfect, they were chosen to ensure we could capture the necessary number of cells for analysis. From a sample of $0.835 \times 10^6$ cells we analysed, there were 168 Pro-HSCs, 244 Pre-HSCs type I and 708 Pre-

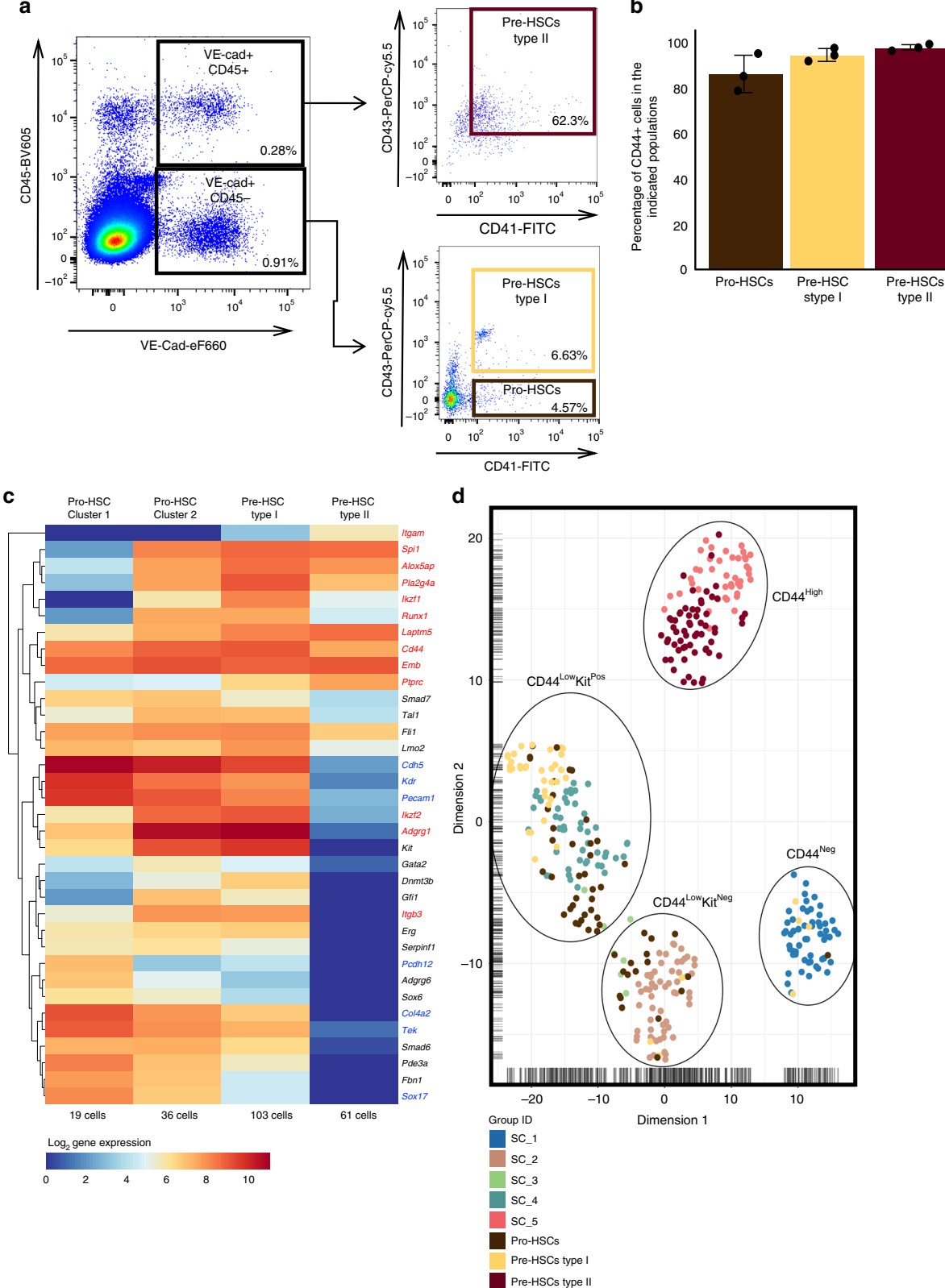

**Fig. 4 Comparison of the CD44 populations with Pro-HSC, Pre-HSC type I and type II. a** FACS plots of VE-Cad, CD45, CD43 and CD41 expression in the AGM region at E10 (31−32 somite pairs). **b** Average of CD44-positive frequency in Pro-HSCs, Pre-HSCs type I and Pre-HSCs type II groups ($n = 3$ independent experiments). Error bars represent SD. Source data are provided as a Source Data file. **c** Single cells from Pro-HSCs (VE-Cad+ CD41+ CD45− CD43−), Pre-HSCs type I (VE-Cad+ CD41+ CD45− CD43+) and Pre-HSCs type II (VE-Cad+ CD45+) populations were isolated and tested by single-cell q-RT-PCR. The heatmap shows the result of the hierarchical clustering analysis (the genes were clustered by Pearson's correlation) based on average expression of indicated gene for each cell cluster. Genes marked in red are blood genes while the ones marked in blue are endothelial genes. **d** tSNE plot combining single-cell q-RT-PCR data from Figs. 3 and 4. See also Supplementary Data 4.

HSCs type II observed. Given these low cell numbers, it was not practical to use more stringent gating.

Interestingly, all populations were CD44 positive (Fig. 4b and Supplementary Fig. 6b). Following hierarchical clustering, we found three clusters: the first mostly composed of Pro-HSCs, a second being a mix of Pro-HSCs and Pre-HSCs type I, and a third one composed only of Pre-HSCs type II (Supplementary Fig. 7 and Fig. 4c). We then performed a clustering analysis in conjunction with the populations defined by CD44 (Fig. 4d). This revealed that the SC_5 (CD44$^{High}$) population closely associated with the Pre-HSC type II and the SC_4 (CD44$^{Low}$Kit$^{Pos}$) population with Pre-HSC type I and part of the Pro-HSCs. SC_2 (CD44$^{Low}$Kit$^{Neg}$) clustered closely with the remaining Pro-HSCs. Finally, only five cells with Pre-HSCs type I and Pro-HSCs phenotype clustered with the SC_1 (CD44$^{Neg}$). Ninety-seven per cent of Pro-HSCs, Pre-HSCs type I and Pre-HSCs type II were CD44 positive (Supplementary Fig. 7a and Supplementary Data 4). Overall, we have demonstrated that the phenotypes based on CD44 expression in combination with Kit could allow us to isolate all key populations in the process of HSCs formation more accurately than the phenotypes previously described.

**CD44$^{Neg}$ and CD44$^{Low}$Kit$^{Neg}$ are distinct transcriptionally**. To further compare the two endothelial populations found in the AGM, we performed RNA-seq on 25-cell bulk samples from these populations across 3 different time points (E9.5, E10 and E11). We analysed as well the more advanced stages in EHT: CD44$^{Low}$Kit$^{Pos}$ at E9.5 and E10, and CD44$^{High}$ at E11. The bulk RNA-seq approach allowed us to detect low abundant genes (such as genes encoding transcription factors) more efficiently than sc-RNA-seq and also to measure smaller changes in gene expression between populations.

The samples clustered according to their marker expression, despite the difference in developmental time, confirming our previous experiments (Fig. 5a and Supplementary Data 5). We identified several haematopoietic genes switched on in the CD44$^{Low}$Kit$^{Neg}$ population, including Ctsc, Nfe2, Runx1 and Ifitm1, suggesting that these cells were subjected to the EHT process (Fig. 5b). The expression pattern of endothelial genes fits with our previous observations—more highly expressed in CD44$^{Neg}$ and CD44$^{Low}$Kit$^{Neg}$, moderately expressed in CD44$^{Low}$Kit$^{Pos}$ and absent in CD44$^{High}$ (Fig. 5b). Moreover, we used this dataset to check the expression of genes corresponding to proteins we found in our antibody screen (Fig. 1a). Cd93 and Madcam1 showed an endothelial expression pattern and could be used to separate endothelial from blood cells in the AGM. In contrast, Itgb3 marks specifically the CD44$^{Low}$Kit$^{Pos}$ population, whereas Ly6e marks both CD44$^{Low}$Kit$^{Pos}$ and CD44$^{High}$ populations as shown previously in Fig. 3 and Supplementary Fig. 5.

Interestingly, this transcriptome analysis showed strong differences between the two endothelial populations of the AGM. We found 1605 genes differentially expressed between these two populations (p-value < 0.01, Wald's test). Among them, several genes from the Notch pathway (Hey2, Jag1, Dll4, Hey1 and Notch1) were significantly more expressed in the CD44$^{Low}$Kit$^{Neg}$ population compared with CD44$^{Neg}$. Similarly, antagonists of the transforming growth factor-β (TGFβ)/bone morphogenetic protein (BMP) pathway, including Smad6, Smad7 and Bmper, were upregulated in CD44$^{Low}$Kit$^{Neg}$ cells compared with CD44$^{Neg}$. In contrast, target genes of the Wnt pathway (Lef1, Ccnd1 and Myc) were more highly expressed in CD44$^{Neg}$ compared with CD44$^{Low}$Kit$^{Neg}$. From this analysis we could also identify specific markers for each of the endothelial populations (Fig. 5c). Nr2f2, Pde2a, Aplnr and Kcne3 marked specifically CD44$^{Neg}$ cells, whereas Adgrg6, Hey2, Akr1c14, Fas and Ltbp1 strongly marked the CD44$^{Low}$Kit$^{Neg}$ population.

The sc-RNA-seq performed by another group investigated the different stages of EHT in the AGM[14], but were unable to identify the CD44$^{Low}$Kit$^{Neg}$ population. The gene expression pattern they obtained for the three other populations was very similar to the one we found with our bulk RNA analysis (Supplementary Fig. 8 and Supplementary Data 6). In addition, a recent study defined the HE in the AGM by the VE-Cad$^+$ Gfi1$^+$ Kit$^-$ phenotype using the Gfi1:H2B-Tomato reporter mouse model[10] and characterized it by sc-RNA-seq[26]. This population matched with the CD44$^{Low}$Kit$^{Neg}$ cells. Indeed, it expressed Cd44, Hey2, Smad6, Smad7 and Adgrg6 but lacked the expression of CD44$^{Neg}$ endothelial cell specific genes such as Pde2a, Aplnr, Nrp2 and Nr2f2 (Supplementary Fig. 9 and Supplementary Data 7).

Of note, a recently published mouse organogenesis sc-RNA-seq atlas[27] allowed us to find that Cd44 gene expression was restricted to a subset of arterial endothelial cells (co-expression of Gja5, Efnb2, Smad6 and Smad7) between E9.5 and E13.5 of mouse embryonic development (Supplementary Fig. 10) supporting our transcriptome results. In contrast, Cd44 expression is absent from most of adult mouse endothelial cells including arterial endothelium in the Tabula Muris sc-RNA-seq atlas[28] (Supplementary Fig. 11).

Overall, our results support the hypothesis that the CD44$^{Neg}$ cells and the CD44$^{Low}$Kit$^{Neg}$ cells belong to two distinct endothelial populations. The CD44$^{Neg}$ population expresses venous (Aplnr, Nr2f2 and Nrp2) and arterial (Sox17, Bmx and Efnb2) markers, whereas the CD44$^{Low}$Kit$^{Neg}$ has a clear arterial signature with stronger expression of Bmx, Jag1 and Hey2. Moreover, there are clear transcriptional links between the two CD44$^{Low}$ populations and the initiation of several blood markers already at the CD44$^{Low}$Kit$^{Neg}$ stage suggests that this population is in fact the endothelial precursor of CD44$^{Low}$Kit$^{Pos}$ and CD44$^{High}$ cells, and hence of haematopoietic development.

**CD44$^{Low}$Kit$^{Neg}$ endothelial cells are quiescent**. A large number of the differentially expressed genes between the two endothelial populations belonged to metabolic processes (395 out of 1605 genes; 1.32-fold enrichment, p-value < 0.05, Fisher's exact test). We therefore identified specific metabolic pathways distinguishing the two populations (Fig. 6a, b and Supplementary Data 8). Notably, the CD44$^{Low}$Kit$^{Neg}$ population showed a pronounced downregulation of genes coding for enzymes involved in glycolysis, the TCA cycle and respiration, suggesting reduced ATP generation. Furthermore, amino acid and nucleotide biosynthesis genes were also downregulated. Altogether, it suggests that the CD44$^{Low}$Kit$^{Neg}$ is in a non-proliferative, metabolically quiescent state, which is in line with the smaller size of this population compared with the CD44$^{Neg}$ (Fig. 2e). To test this hypothesis, we performed a cell cycle analysis of the four different CD44 populations at E10 and we found that indeed CD44$^{Low}$Kit$^{Neg}$ population was the most quiescent compared with the other endothelial subset (Fig. 6c and Supplementary Fig. 12). Interestingly, the proliferation rate increased, as the cells were progressing through EHT consistent with the observations made by another group[29].

Given that endothelial cells are known to obtain most of their energy from glycolysis[30], this change in metabolic status suggests a loss of endothelial identity. These cells also show a marked increase in the expression of pathways leading to lipids with regulatory function: glycerolipids, glycerophospholipids, phosphatidylinositol and sphingolipids (Supplementary Fig. 13a).

Moreover, we observed that several genes involved in autophagy, a process known to be regulated by phosphatidylinositol and sphingolipids[31–34], were also highly upregulated in the CD44$^{Low}$Kit$^{Neg}$ population (Supplementary Fig. 13b). Concordantly, two key

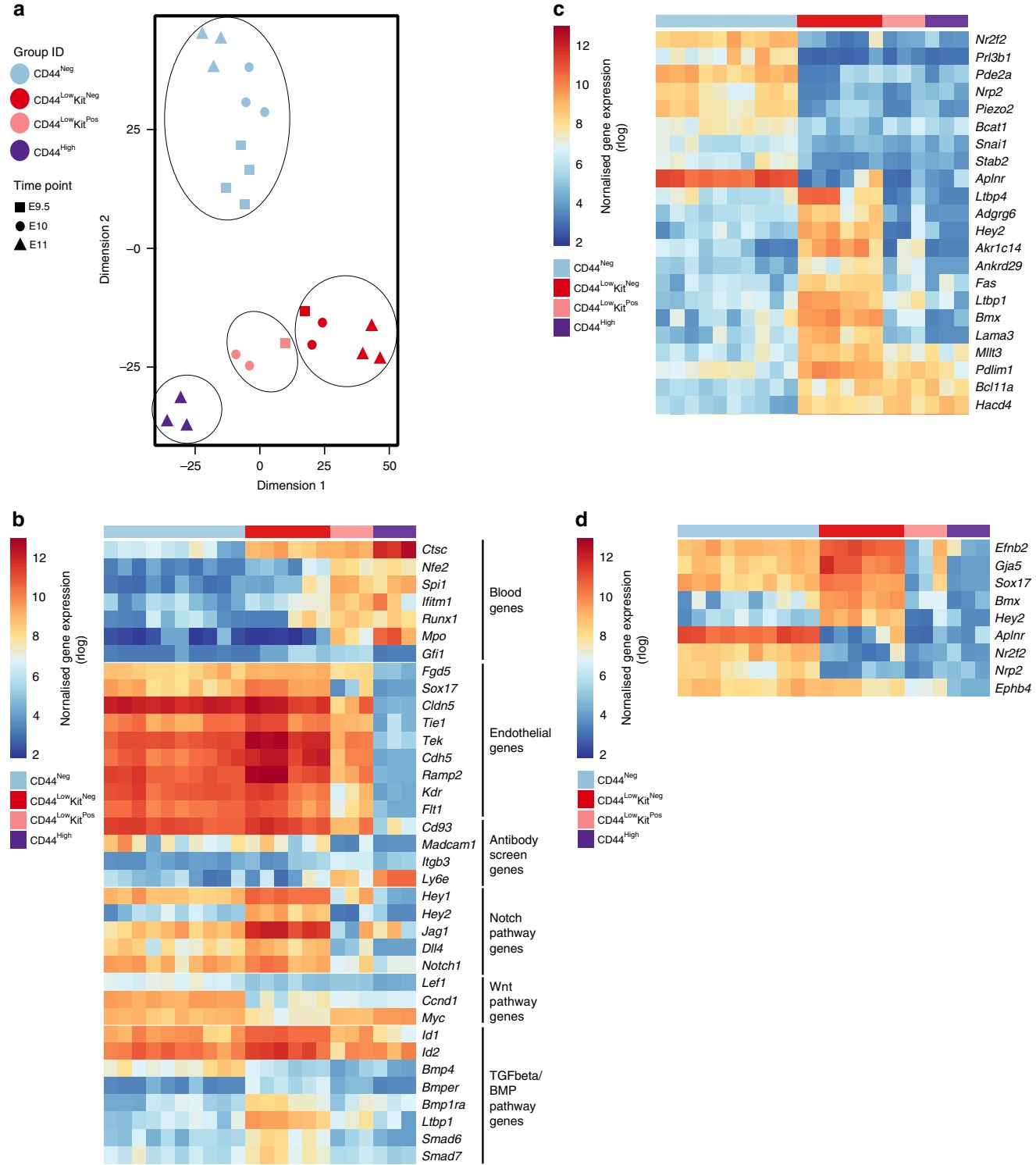

**Fig. 5 Bulk RNA sequencing identifies early changes in the EHT process. a** tSNE plot from 25-cell bulk RNA sequencing generated from E9.5, E10 and E11 AGM. Four groups are indicated. **b** Heatmap of gene expression highlighting a selection of genes. **c** Heatmap of gene expression highlighting the most differentially expressed genes between CD44Neg and CD44LowKitNeg endothelial populations (*p*-value < 0.01). **d** Heatmap of gene expression highlighting arterial (*Efbn2*, *Gja5*, *Sox17*, *Bmx* and *Hey2*) and venous (*Aplnr*, *Nr2f2*, *Nrp2* and *Ephb4*) coding genes in CD44LowKitNeg and CD44Neg populations. See also Supplementary Data 5.

processes accompanying autophagy, ubiquitylation and proteolysis, were also upregulated. As autophagy has been shown to play a key role in embryonic development and haematopoiesis[35–37], this provides further support to the CD44LowKitNeg cells being in transit from endothelial to haematopoietic identity.

**Runx1 is dispensable for the formation of CD44LowKitNeg.** CD44 has allowed us to clearly define the key VE-Cad+ populations in the AGM. The transcription factor RUNX1 is essential for HSPC formation and is known to downregulate endothelial identity through its target genes *Gfi1* and *Gfi1b*[10,38].

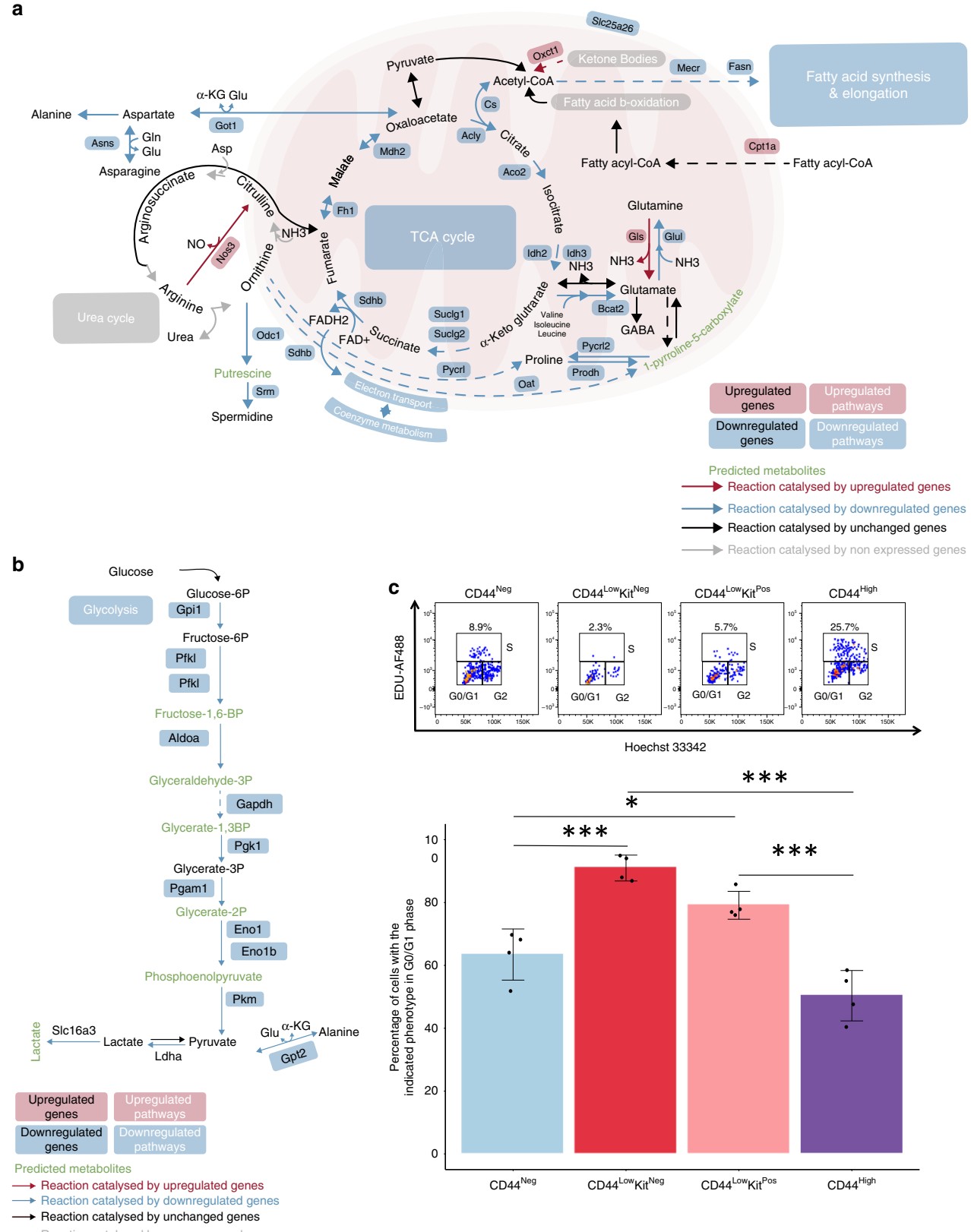

Next, we decided to evaluate the impact of *Runx1* loss of function on the different CD44+ cells. Using a *Runx1* knockout *mouse* model[39], we stained for VE-Cad, CD44 and Kit expression, and performed transcriptional profiling on the sorted cells (Fig. 7, Supplementary Fig. 14 and Supplementary Data 9). We found that in the absence of *Runx1* there is a loss of CD44^High and CD44^LowKit^Pos cells, and we observed a concomitant increase in the frequency of the CD44^LowKit^Neg population (Fig. 7a, b). Interestingly, we found no obvious transcriptional differences between the CD44^LowKit^Neg populations derived

**Fig. 6 Reduced expression of genes involved in TCA cycle and glycolysis in CD44$^{Low}$Kit$^{Neg}$ endothelial cells. a**, **b** Overview of key metabolic nodes and pathways enriched in differentially expressed genes when comparing the CD44$^{Low}$Kit$^{Neg}$ and CD44$^{Neg}$ endothelial populations. These were selected based on reporter metabolite analysis. Pathway boxes summarize multiple genes/reporter metabolites (Supplementary Data 8). The mentioned upregulated and downregulated genes refer to the expression in the CD44$^{Low}$Kit$^{Neg}$ population compared with CD44$^{Neg}$. **c** Cell proliferation analysis of the four indicated populations showing a representative flow cytometry profile of cell cycle status (top panel) and a bar graph summarizing the proportion of cells in G0/G1 cell cycle phases for each population (bottom panel) ($n = 4$ independent experiments). Error bars represent SD. A one-way ANOVA was used to compare the proportion of cells in the G0/G1 phase across all populations $F(3,12) = 30.24$, $p$-value < 0.0001. A Tukey's HSD post-hoc test was used to determine where the significance lies, *$p$-value < 0.05, **$p$-value < 0.01, ***$p$-value < 0.001. Source data are provided as a Source Data file.

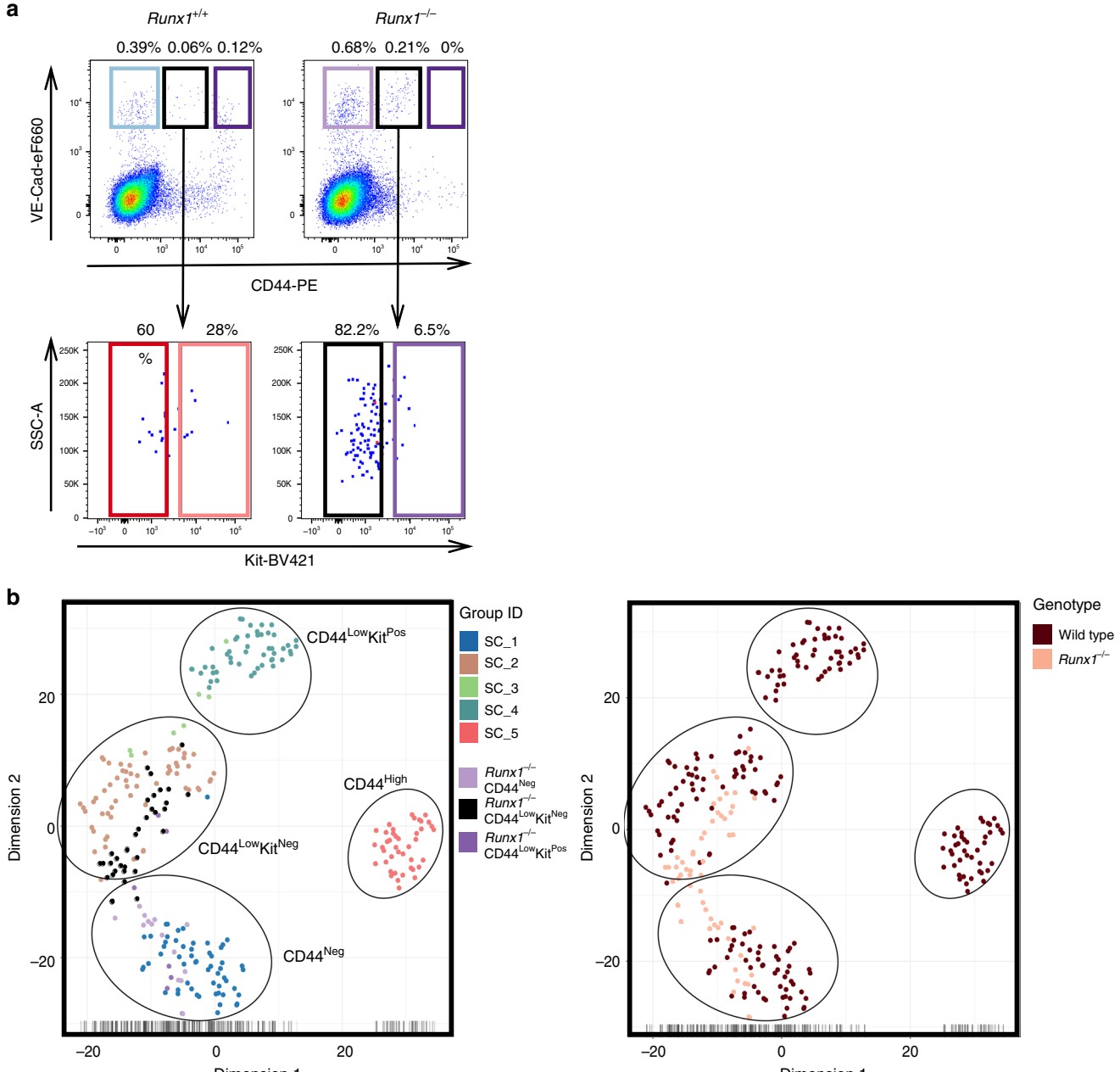

**Fig. 7 Runx1 is required for the generation of CD44$^{Low}$Kit$^{Pos}$ and CD44$^{High}$ cells. a** FACS plots of VE-Cad and CD44 expression in the AGM region at E10.5 from Runx1$^{+/+}$ (left) and Runx1$^{-/-}$ (right) embryos. **b** tSNE plots from single-cell q-RT-PCR data shown in **a**. See also Supplementary Fig. 14 and Supplementary Data 9.

from Runx1$^{+/+}$ vs. Runx1$^{-/-}$ embryos, indicating that RUNX1 is not necessary for the formation of these endothelial cells but for the promotion of the transition into CD44$^{Low}$Kit$^{Pos}$ cells (Supplementary Fig. 14).

**VE-Cad$^{Pos}$ CD44$^{Pos}$ cells display haematopoietic potential.** To understand the haematopoietic potential of the different populations defined by VE-Cad, CD44 and Kit expression, we performed ex vivo assays using an OP9 co-culture system. No

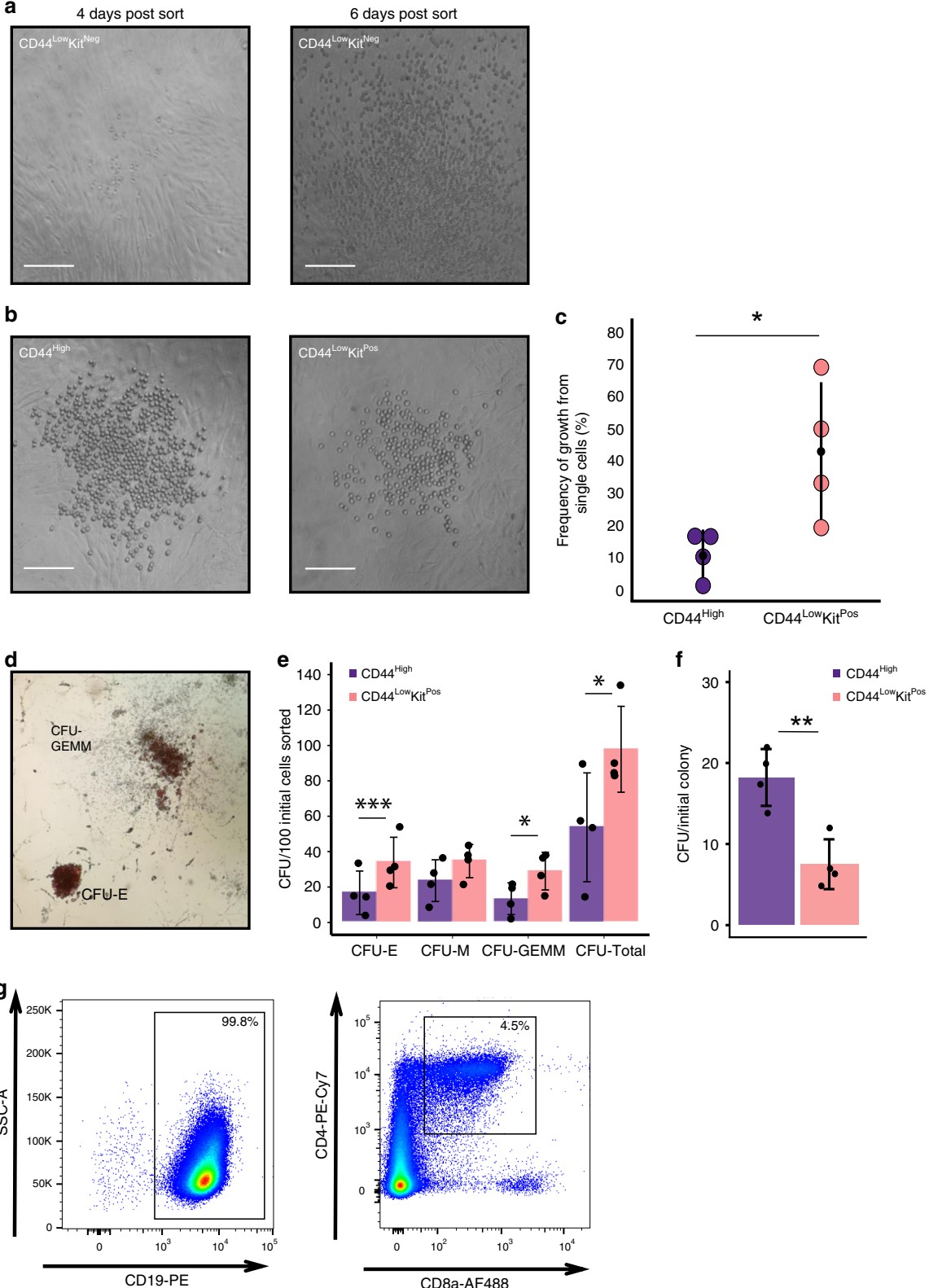

colonies were formed at the single-cell level from either the CD44$^{Low}$Kit$^{Neg}$ or the CD44$^{Neg}$ populations. However, by plating cells at a density of 300 cells per well, we could observe round cell colony formation from the CD44$^{Low}$Kit$^{Neg}$ population (4 out of 16 wells) but not from the CD44$^{Neg}$ (0 out of 13 wells) (Fig. 8a). As this growth occurred 4 times out of a total of 16 wells, the frequency of cells giving rise to blood in CD44$^{Low}$Kit$^{Neg}$

population is <1 in 300, which is in line with the very low expression of haematopoietic genes in this population.

In contrast, using single-cell sorting, we found that the CD44$^{Low}$Kit$^{Pos}$ population was the most potent with an average of 43% of single cells forming round cell colonies after 3 days of growth. Similarly, the CD44$^{High}$ population produced haematopoietic colonies but with a lesser frequency; on average, 11% of

**Fig. 8 VE-Cad$^+$ CD44$^+$ populations have haematopoietic potential. a** Images of OP9 co-cultures after 4 and 6 days of incubation. Haematopoietic potential was observed from CD44$^{Low}$Kit$^{Neg}$ cells with colonies of round cells resulting from 300 cells sorted per well. No round cell colonies were observed with CD44$^{Neg}$ cells. Scale bars represent 100 μM. **b** Images of OP9 co-culture after 3 days of culture are shown. A single CD44$^{High}$ or CD44$^{Low}$Kit$^{Pos}$ cell was FACS sorted onto a confluent OP9 stromal layer and incubated in HE medium. Scale bars represent 100 μM. **c** The percentage of single cells giving rise to colonies was quantified across four independent experiments ($n = 4$). The graph compares the frequency of growth from single CD44$^{High}$ ($M = 0.11$, SD $= 0.07$) cells with CD44$^{Low}$Kit$^{Pos}$ cells ($M = 0.43$, SD $= 0.22$). Statistical significance was determined by a two-tailed, paired $t$-test, $t(3) = -4.11$, $p$-value $=$ 0.026. Source data are provided as a Source Data file. **d** Colony-forming unit assays were performed following three days OP9 culture of CD44$^{Low}$Kit$^{Pos}$ and CD44$^{High}$ cells (100 cells per well). Cells were kept for a further 7 days in methocult medium before quantification. Images show representative CFU-E and CFU-GEMM colonies. **e** The bar graphs show the number of CFUs generated per 100 initial FACS sorted cells ($n = 4$ independent experiments). Significance was determined by two-tailed, paired $t$-tests (see Source Data file). **f** The bar graph indicates the total number of colony-forming units formed per initial colony grown on the OP9 stromal layer. Although the CD44$^{Low}$Kit$^{Pos}$ population gives rise to approximately six times more round cell colonies on OP9 then the CD44$^{High}$ population, only ~2.5 times more CFUs are generated. Significance was determined by a two-tailed, paired $t$-test $t(3) = 6.67$, $p$-value $= 0.0069$ ($n = 4$ independent experiments). Source data are provided as a Source Data file. **g** B- and T-cell lymphoid assays were performed following 21 days of OP9 (B cells) and OP9-DL1 (T cells) culture of 50 FACS-sorted CD44$^{High}$ cells. Percentages of CD19$^+$ (B-cells) and CD4$^+$CD8a$^+$ (immature T-cells) are shown. *$p$-value $< 0.05$, **$p$-value $< 0.01$, ***$p$-value $< 0.001$. Error bars indicate SD.

single cells showed the ability to form haematopoietic colonies on OP9 (Fig. 8b, c). To uncover the differentiation potential of the cells generated by CD44$^{Low}$Kit$^{Pos}$ and CD44$^{High}$, we performed colony-forming unit (CFU) assays following the OP9 co-culture. Both populations readily generated both erythroid and myeloid colonies with the CD44$^{Low}$Kit$^{Pos}$ population demonstrating a significantly higher capacity than the CD44$^{High}$ population (Fig. 8d, e). However, the fourfold increase in the number of CD44$^{Low}$Kit$^{Pos}$ colonies on OP9 did not correspond to a fourfold increase in CFU colonies, suggesting a higher replating efficiency of the CD44$^{High}$ cells compared with CD44$^{Low}$Kit$^{Pos}$ (Fig. 8f). We further tested the lymphoid potential of the CD44$^{High}$ population by growing cells for 21 days on either OP9 or OP9-DL1 with lymphoid-promoting cytokines. We demonstrated that this population could give rise to both B and T cells ex vivo (Fig. 8g and Supplementary Fig. 15).

Our transcriptome and fluorescence-activated cell sorting (FACS) analyses showed that CD44$^{Low}$Kit$^{Pos}$ were similar to the Pre-HSC type I, whereas the CD44$^{High}$ were equivalent to Pre-HSC type II (VE-Cad$^{Pos}$ CD45$^{Pos}$). Pre-HSC type I gives rise to type II in OP9 co-culture[40] and, as expected, by isolating CD44$^{Low}$Kit$^{Pos}$ cells, we could demonstrate that they were producing VE-Cad$^{Pos}$ CD45$^{Pos}$ on their way to produce blood cells supporting the differentiation relationship between the two CD44$^{Pos}$ Kit$^{Pos}$ populations (Supplementary Fig. 16).

Overall, we found that all populations expressing CD44 displayed the capacity to produce haematopoietic cells on OP9 cells including CD44$^{Low}$Kit$^{Neg}$ albeit at a very low frequency.

**Interrupting hyaluronan binding to the CD44 reduces EHT.** So far, we demonstrated that CD44 was a very useful marker to distinguish the different stages of EHT. Although the *Cd44* knockout mice do not have a severe haematopoietic phenotype[41], compensatory mechanisms through other Hyaluronan receptors (e.g., *Hmmr*[42] expressed by some CD44$^{Low}$Kit$^{Pos}$ and CD44$^{High}$ cells in Fig. 3a) may be at play to diminish the consequences of CD44 loss of function. To explore the functional role of CD44 in EHT, we employed a pharmacological approach. By treating CD44$^{High}$-sorted cells with a CD44 blocking antibody known to bind close to the hyaluronan binding site on the extracellular domain of CD44[29], we found that round cell colony formation could be inhibited in a dose dependent manner (Fig. 9a, b). The blocking antibody inhibited not only the number of colonies deriving from the ex vivo sorted cells but also the size of the colonies generated (Fig. 9c).

To further investigate the role of CD44 in EHT, we used the in vitro ESC differentiation system mimicking embryonic haematopoiesis[24]. We performed a haemangioblast culture and

analysed CD44 expression at day 1, 2 and 3. Endothelial cells expressed CD44 at all tested time points (Fig. 9d).

We therefore applied the blocking antibody from the beginning of the culture and found that treatment with the antibody halved the number of HSPCs (VE-Cad$^-$ CD41$^+$) produced and significantly increased the percentage of endothelial cells in the culture (Fig. 9e, f). We could not get this effect using another CD44 antibody, which did not bind close to the hyaluronan binding site (Supplementary Fig. 17), suggesting the importance of CD44 interaction with its ligand for EHT. Given this result, we next attempted to manipulate the amount of hyaluronan in the culture. When 300 μg/mL of hyaluronidase was applied to the haemangioblast culture, we again observed a block in EHT characterized by a significant decrease in blood cell formation and an increase in the percentage of Pre-HSPCs (VE-Cad$^+$ CD41$^+$) (Fig. 9f). Using the methylumbelliferone (4MU) inhibitor, which blocks the synthesis of hyaluronan, we also observed the reduction of CD41$^+$ cell number. Combining 4MU with hyaluronidase had a much more potent effect with a clear block in EHT at the Pre-HSPC stage.

In conclusion, these results demonstrate a regulatory role for hyaluronan and its receptor CD44 in the formation of HSPCs.

## Discussion

Using antibody screening and sc-RNA-seq, we discovered that CD44 was a robust marker to distinguish all the main populations in the EHT process in the AGM. Combining CD44 with Kit and VE-Cad allowed us to discriminate the different stages of EHT more accurately than the method based on VE-Cad, CD41, CD43 and CD45 cell surface markers[13]. In addition, we showed that CD44 had an unexpected regulatory function in the EHT process.

Our work has been instrumental in distinguishing the different types of endothelial cells in the AGM region. Previously, Zhou et al.[14] found only one type of AGM endothelial cells, identical to the CD44$^{Neg}$ population we described in the present study (Supplementary Fig. 8). On the other hand, the study by Baron et al.[26] described two endothelial populations: non-HE and HE. Surprisingly, they had no more than 100 genes differentially expressed between them. In contrast, we found 1605 genes changing between the two AGM endothelial populations we identified. A close look at their E11 sc-RNA-seq data showed that their non-HE cells were heterogeneous. Seventy-three per cent of them were in fact HE cells with characteristics identical to the CD44$^{Low}$Kit$^{Neg}$ population, while the remaining cells formed a separate cluster of CD44$^{Neg}$ cells (Supplementary Fig. 9). Therefore, our study is the first one to identify and compare thoroughly the different types of AGM endothelial cells. The

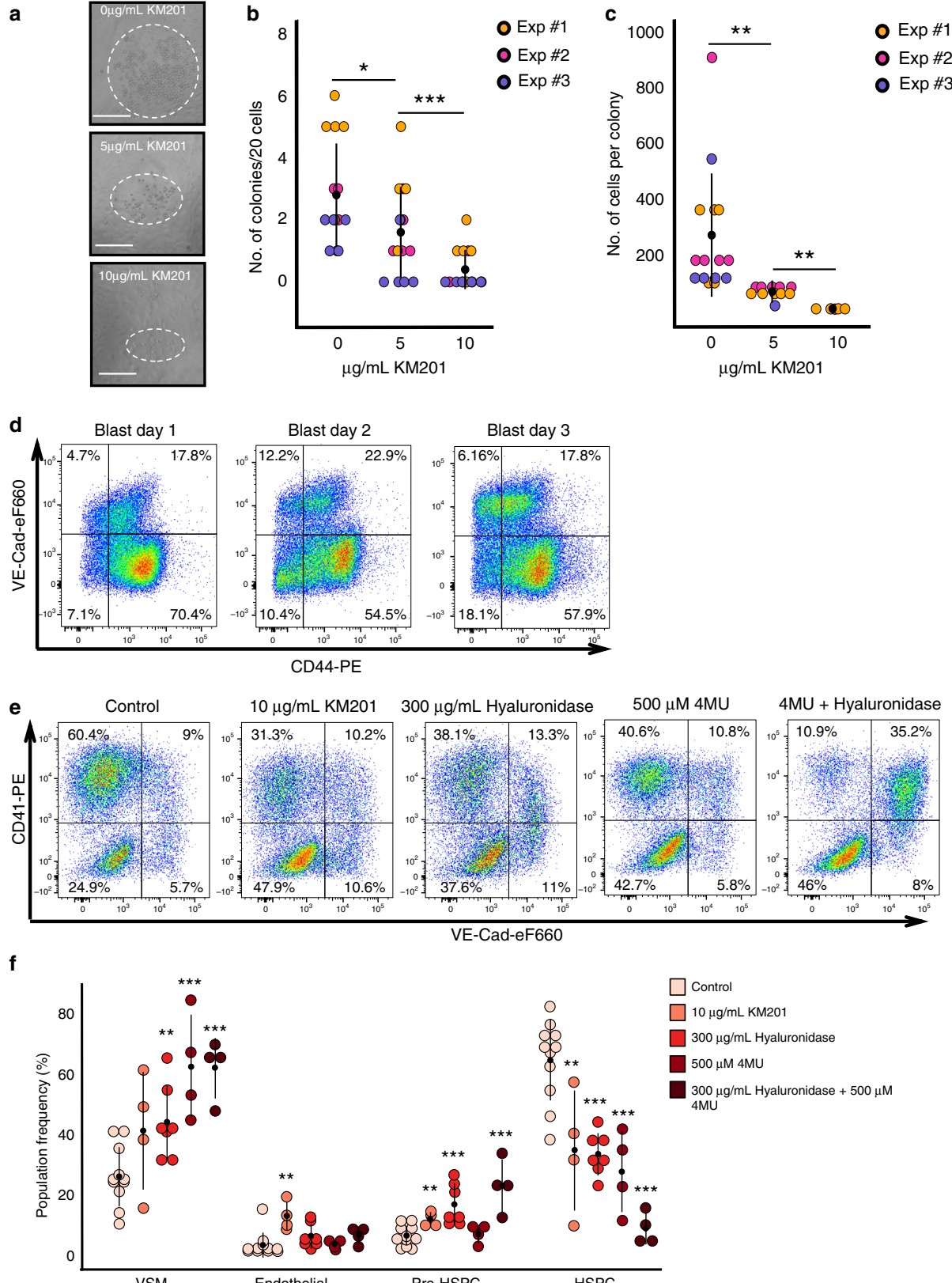

CD44[Low]Kit[Neg] cells have a gene expression signature strongly compatible with arterial identity (e.g., expression of *Efbn2* and *Sox17*, and upregulation of Notch pathway target *Hey2*), while the CD44[Neg] population co-expressed genes related to the venous (e.g., *Nr2f2* and *Aplnr*) and arterial cell fates (e.g., *Efnb2* and *Sox17*) (Fig. 5d). It is in line with previous sc-RNA-seq datasets from the AGM[14,26]. In the first analysis by Zhou et al.[14], 64% of CD44[Neg] cells (18 out of 28) were *Efnb2+Aplnr+*, whereas 50% of them (14 out of 28) were *Sox17+Aplnr+* (Supplementary Fig. 8). In the second dataset by Baron et al.[26], 36% of them (4 out of 11) were *Efnb2+Aplnr+* and 18% (2 out of 11) were positive for *Sox17* and *Aplnr* (Supplementary Fig. 9).

**Fig. 9 Blocking the interaction between CD44 and hyaluronan inhibits the EHT. a** Images of round cell colonies generated from CD44[High] cells after 4 days of OP9 co-culture with different concentrations of KM201 anti-CD44 blocking antibody. Dotted line indicates approximate size of colonies. Scale bar corresponds to 100 μm. **b** Dot plot comparing number of round cells colonies formed as a function of the concentration of anti-CD44 blocking antibody applied. Source data are provided as a Source Data file. **c** Dot plot indicating the number of cells per colony as a function of the concentration of anti-CD44 blocking antibody applied. **b, c** Kruskal–Wallis tests were used to compare treatment groups and Dunn post-hoc test with Benjamini–Hochberg adjustment was used to identify significance between the conditions (see Source Data file) where *$p$-value < 0.05, **$p$-value < 0.01 and ***$p$-value < 0.001. Each data point represents one well. For each of the three independent experiments ($n = 3$), there were five different wells. Source data are provided as a Source Data file. **d** Flow cytometry analysis of CD44 and VE-Cad expression in Hemangioblast culture between day 1 and day 3. The dot plots show expression of VE-Cadherin and CD44 at the indicated time points. **e** Representative FACS plots of VE-Cad and CD41 expression after 2 days of haemangioblast differentiation. Cells were either untreated (control) or treated with anti-CD44 blocking antibody, hyaluronidase enzyme, 4MU hyaluronan synthase inhibitor or a combination. **f** Dot plot showing the population percentage for vascular smooth muscle (VSM) (VE-Cad[−]CD41[−]), endothelial cells (VE-Cad[+]CD41[−]), Pre-HSPCs (VE-Cad[+]CD41[+]) and HSPCs (VE-Cad[−]CD41[+]) after 2 days of haemangioblast differentiation, summarizing the results of FACS analysis shown in **d**. Each data point represents an independent experiment for the indicated conditions ($n = 11$ for control, $n = 4$ for 10 μg/mL KM201, $n = 7$ for 300 μg/mL Hyaluronidase, $n = 4$ for 500 μM 4MU and $n = 4$ for 300 μg/mL Hyaluronidase + 500 μM 4MU). Significance was determined for VSM, pre-HSPCs, and HSPC populations using a one-way ANOVA followed by Dunnett's post-hoc tests (see Supplementary Material). As the distribution of values for the endothelial population was not normally distributed a Kruskal–Wallis test was applied and significant differences evaluated with a Dunn post-hoc test (see Source Data file). For **d**, *$p$-value < 0.05, **$p$-value < 0.01 and ***$p$-value < 0.001. Error bars represent SD. Source data are provided as a Source Data file.

Of note, not all of the CD44[Neg] cells co-expressed arterial and venous markers, suggesting heterogeneity within this population (Supplementary Figs. 8 and 9). This could be attributed to the fact that some of these endothelial cells come from the dorsal aorta (Supplementary Fig. 3) and the rest from the cardinal veins, which are entirely CD44[Neg] (Supplementary Fig. 10). Indeed, cardinal veins are not removed when the AGM region is isolated, because they are very close to the aorta. It would explain the presence of endothelial cells expressing only venous markers within the CD44[Neg] population.

Another interesting finding was the striking metabolic state difference between the two endothelial populations. The CD44[Neg] cells appeared much more metabolically active than the CD44[Low]Kit[Neg], suggesting that the latter population is in a state of quiescence. This was further supported by our cell cycle analysis (Fig. 6). This is surprising given that the acquisition of a quiescent phenotype in endothelial cells occurs normally after birth following the completion of angiogenesis. FOXO1 has been described as an important transcription factor to induce quiescence in endothelial cells through suppression of Myc[43]. Although we observe specific downregulation of Myc in CD44[Low]Kit[Neg] cells, *Foxo1* has a similar level of expression in the two endothelial populations (Supplementary Data 5). Recently, a study comparing lung endothelial cells between infant and adult mice showed that higher expression of SMAD6 and SMAD7 was linked with the induction of endothelial cell quiescence typical in adulthood[44]. Interestingly, all the CD44[Low]Kit[Neg] cells co-expressed *Smad6* and *Smad7* at the single-cell level, suggesting that the quiescent phenotype we observe could also be linked to the co-expression of the two inhibitory Smads.

The acquisition of quiescence in this context could be the first indication of haemogenic capacity. The metabolic features of the CD44[Low]Kit[Neg] population were coupled with an increase in genes involved in autophagy. This is in line with the known role of autophagy in embryogenesis, haematopoiesis and stem cell maintenance[35–37,45,46], such as protein and organelle turnover and protection from reactive oxygen species. Our data thus suggest that these cells mark the resetting of metabolic and regulatory states to initiate the EHT process.

We consequently propose that the CD44[Low]Kit[Neg] population is the source of HSPCs in the AGM (Fig. 10). Although it was suggested that HE cells in the human pluripotent stem cell differentiation model do not display arterial identity[47], our results clearly support the arterial source of HSPCs in the AGM. Of note, the CD44 protein has been detected in intra-aortic

haematopoietic clusters during human embryogenesis[48], suggesting an evolutionary conservation of its expression pattern.

Another characteristic for the initiation of the EHT is the inhibition of the BMP and TGFβ pathways as suggested by the increased expression of *Smad6*, *Smad7* and *Bmper* in the CD44[Low] Kit[Neg] arterial population. Expression of RUNX1 in endothelial cells would then trigger haematopoietic gene expression. The dynamic interaction between the heptad of transcription factors GATA2, RUNX1, LYL1, ERG, FLI1, LMO2 and TAL1 co-expressed at the Pre-HSPC-I stage would eventually lead to the loss of endothelial gene expression[24] and give rise to the Pre-HSPC-II stage, cells expressing only haematopoietic genes.

Interestingly, we also detected CD44 on endothelial cells generated in the in vitro EHT model based on ESC differentiation. Of note, the analysis of the mouse organogenesis sc-RNA-seq atlas[27] showed that the *Cd44* gene was mostly expressed by a fraction of arterial endothelial cells indicating that this gene has a restricted expression pattern during development (Supplementary Fig. 10). It made CD44 expression on endothelial cells in this in vitro model even more significant. In addition, we have shown that endothelial cells produced in vitro co-express the *Smad6* and *Smad7* genes as in the AGM CD44[Low] Kit[Neg] population[24,49]. The production of blood progenitors in vitro is also going through an intermediary stage expressing endothelial, haematopoietic genes, the key transcription factors *Erg*, *Fli1*, *Lmo2*, *Lyl1*, *Fli1*, *Tal1*, *Runx1* and *Gfi1* as in the AGM[24]. Finally, even though HSCs have not yet been generated from ESCs without the use of reprogramming, definitive erythroid, myeloid and lymphoid cells could be obtained from this model[50]. These lineages could also be produced upon transplantation of in vitro derived progenitors in adult mice and could be detected up to 22 weeks after injection, indicating a long-term reconstitution[50]. These elements convinced us that studying CD44 role in the in vitro haemangioblast culture would be relevant to AGM EHT.

Our detailed transcriptomics analysis of haemogenic and non-haemogenic endothelial cell populations in the AGM revealed the prerequisites needed by endothelial cells to generate HSPCs. Thus, our study may help to design protocols for the generation of HSCs ex vivo without the use of transcription factors, which still represents a considerable health risk. The manipulation of the interaction between CD44 and hyaluronan could offer a strategy for reprogramming endothelial cells into HSPCs.

In addition, our work could shed light on CD44 function in other cellular transitions such as the epithelial–mesenchymal

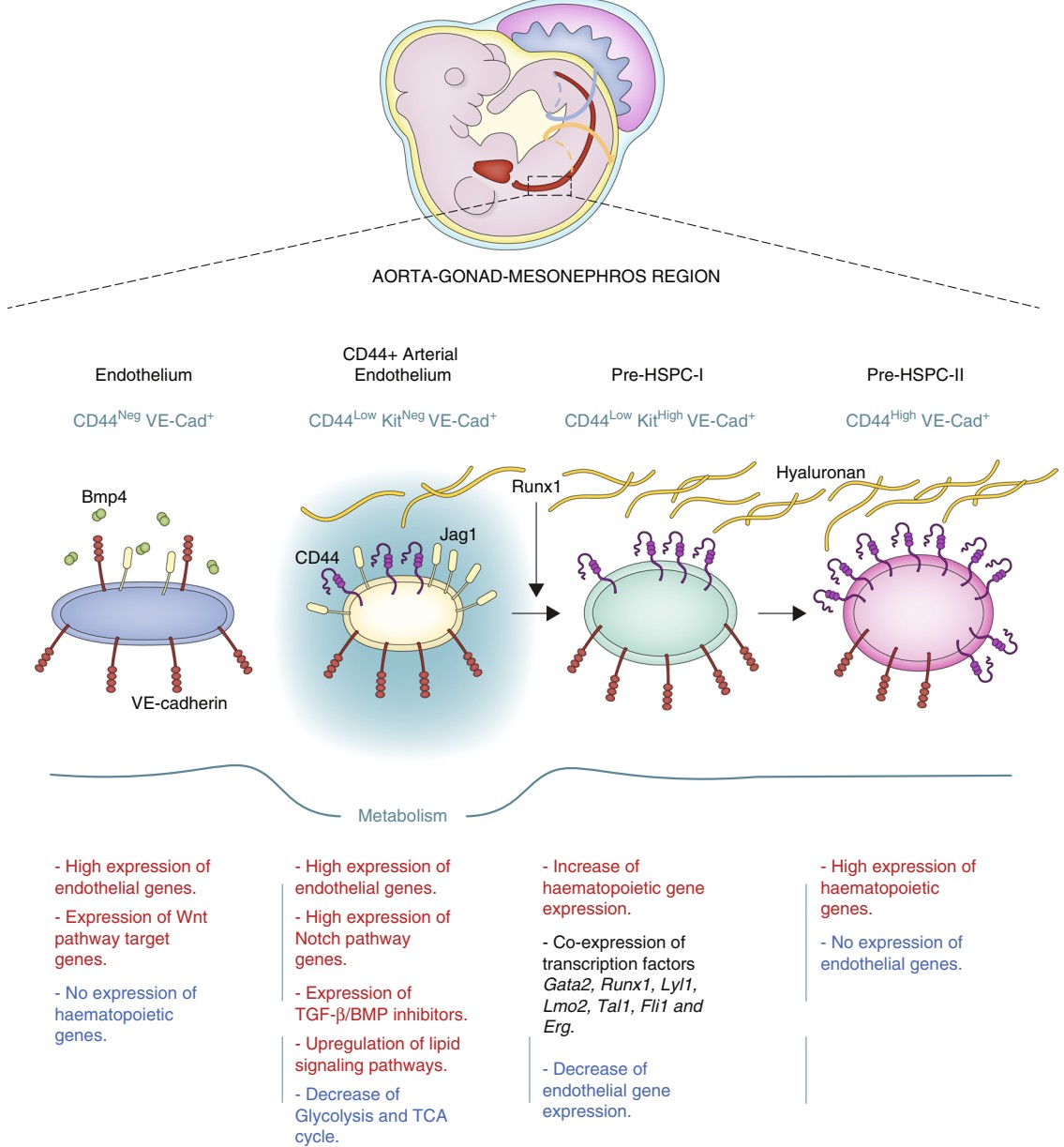

**Fig. 10 New model for the progression of EHT.** Scheme summarising the findings of the present study. We identified two different sets of endothelial cell populations in the AGM with very distinct properties in term of signalling pathways and metabolic states. Expression of Runx1 in the CD44+ arterial endothelial population triggers the upregulation of haematopoietic genes and the formation of the Pre-HSPC-I, which co-expresses endothelial and haematopoietic genes. Continuous expression of haematopoietic genes and interaction between CD44 and hyaluronan eventually lead to the loss of endothelial genes and the formation of Pre-HSPC-II expressing only haematopoietic genes.

transition occurring during metastasis[23]. Given the role of CD44 in the metastatic process, it is possible that there is an overlap in its function for these transformations. Therefore, understanding the down-stream targets of the CD44–hyaluronan interaction could also have implications for cancer biology.

## Methods

**Timed mating and embryo dissection**. For timed pregnancies, C57BL/6 wild-type mice or Runx1[+/−] mice (two to three months old) were mated overnight. The welfare of adult mice used in this work was covered by the licence number 17/2019-PR approved by the Italian Health Ministry. Embryos were collected in phosphate-buffered saline (PBS) supplemented with 10% fetal bovine serum (FBS) (PAA Laboratories). The yolk sac of embryos derived from Runx1[+/−] mice were genotyped using the Kappa *Mouse* Genotyping Kit (KAPA Biosystems) according to the manufacturer's instructions. E9.5 to E11.5 embryos were staged based on somite counts. After removing the yolk sac, the AGM region was dissected by

removing the head and tail above and below the limb buds then removing the limb buds, organs and somites by cutting ventrally and dorsally either side of the AGM.

All experiments were performed following the guidelines and regulations defined by the European and Italian legislations (Directive 2010/63/EU and DLGS 26/2014, respectively). They apply to foetal forms of mammals as from the last third of their normal development (from day 14 of gestation in the mouse). They do not cover experiments done with day 12 mouse embryos and at earlier stages. Therefore, no experimental protocol or license was necessary for the experiments performed on mouse embryos. Mice were bred and maintained at the EMBL Rome Animal Facility in accordance with European and Italian legislations (EU Directive 634/2010 and DLGS 26/2014, respectively).

**C1 Fluidigm chip and single-cell RNA sequencing**. AGM from E10 embryos were dissected and dissociated with collagenase (Sigma) at 37 °C for 30 min. The collagenase was deactivated with PBS supplemented with 10% FBS (Gibco). Cells were stained with anti-VE-Cad and anti-CD41 antibodies (Supplementary Table 3). VE-Cad[+] cells from E10 embryos were isolated using FACS and mixed with Fluidigm suspension reagent (Fluidigm) in a 3:2 ratio. A primed Fluidigm C1 chip and 5 µl

of cell suspension was loaded onto a C1 instrument. Cell capture was then assessed with a ×40 bright-field microscope and wells scored as single cell, doublet or debris. Lysis, reverse transcription and PCR reagents (Clontech Takara) along with ERCC spike-ins at 1 in 4000 dilution (Ambion) were added and mRNA-Seq RT & Amp script were performed overnight. The cDNA was then collected and diluted in Fluidigm C1 DNA dilution buffer.

**Single-cell RNA sequencing data analysis**. The paired 2 ×101 bp Illumina reads from the libraries were quantified using Salmon[51] with the setting -l IU to indicate library topology and the optional flags–posBias and –gcBias, to account for coverage and amplification biases present in sc-RNA-seq protocols. As an index the cDNA annotation of Ensembl release 85 for GRCm38.p4 was used, together with ERCC spike-in sequences. The transcript per million (TPM) values were re-scaled to not include ERCC expression and only consider endogenous gene expression.

Technical features of the data were compared with manual gene annotation of samples in C1 chambers through microscopy. Samples with less than 500,000 mapped reads and >30% mitochondrial content were discarded from analysis. This left 78 single cells in the in vivo experiment.

To identify cells as endothelial or haematopoietic, expression levels of ten known markers were analysed (*Cdh5, Kdr, Pecam1, Pcdh12, Sox7, Gfi1, Gfi1b, Myb, Runx1, Spi1*). Cells were clustered using Principal Component Analysis (PCA) and a Gaussian Mixture Model with two components on these markers (Fig. 1b). A cluster of ten cells considered haematopoietic was identified (and annotated based on high Runx1 expression). Analysis was performed using the decomposition.PCA and mixture.GaussianMixture classes in the scikit-learn package. PCA analysis was performed on the log-transformed TPM values of the markers and the first two principal components were used for Gaussian Mixture.

Markers for hematopoietic cells were identified using a likelihood ratio test, where the alternative model included a binary term for whether the cell was haematopoietic, and the null model just assumed a common mean for all the cells. The *P*-values from the likelihood ratio test were corrected for multiple testing by the Bonferroni procedure. The top differentially expressed genes were investigated to find markers, which could be used in follow-up experiments, and *Cd44* was considered a good candidate (Fig. 1c).

**Analysis of the mouse single-cell transcriptome atlases**. Seurat v.2 was used to perform sc-RNA-seq clustering analysis on the data retrieved from the *Mouse* Organogenesis Cell Atlas[27]. We took the filtered data cds_cleaned.RDS from which the doublet and low-mRNA cells were removed. For annotation and metadata, the cell_annotation.csv table was used.

We selected the cells related to the Endothelial trajectory. For these files we created Seurat object and ran clustering analysis including normalization with LogNormalize method. Afterwards, we found 4890 variable genes using the following parameters: x.low.cutoff = 0.0125, x.high.cutoff = 3, y.cutoff = 0.5. Next, we performed a scaling with linear model regressed on the number of UMI and clustering with dimensions 1:20 and resolution 1.0. With clustered data we generated t-SNE plots highlighting the genes we were interested in (Supplementary Fig. 10).

The Tabula Muris[28] Gene-count tables for FACS sorted adult Aorta, Brain, Diaphragm, Fat, Heart, Kidney, Limb, Liver, Lung, Mammary Gland, Pancreas and Trachea cells were downloaded from Figshare: https://figshare.com/articles/Single-cell_RNA-seq_data_from_Smart-seq2_sequencing_of_FACS_sorted_cells_v2_/5829687.

The expression matrix was then processed through the CONCLUS pipeline[52]. https://github.com/lancrinlab/CONCLUS. We then generated t-SNE plots highlighting the expression pattern of *Cdh5* and *Cd44* genes (Supplementary Fig. 11).

**In vitro ES cell differentiation system**. The A2lox Empty embryonic stem (ES) cell line[24] was maintained and differentiated as follows. ES cells were maintained on mouse embryonic fibroblasts (MEFs) in KnockOut™ Dulbecco's modified Eagle's medium (DMEM) medium (Gibco, 10829018) supplemented with 1% Penicillin–Streptomycin (Gibco, 15070063), 1% L-glutamine (Gibco, 25030081), 1% MEM non-essential amino acids solution (Gibco, 11140050), 15% FBS (Gibco, 10270106), 0.024 μg mL⁻¹ of LIF (EMBL Protein expression and purification core facility) and 0.12 mM of 2-Mercaptoethanol (Gibco). Cells were maintained on MEFs and incubated at 37 °C with 5% CO₂ and 95% relative humidity. TrypLE™ Express Enzyme (1×) (Gibco, 12605036) was used to detach cells for passaging, collection or to create a single-cell suspension.

To begin differentiation into haemangioblast, the ESCs were passaged twice on 0.1% gelatin to remove MEFs first in DMEM-ES medium and then using IMDM-ES medium containing IMDM (Iscove's modified Dulbecco medium) (Lonza; BE12-726F) supplemented with 1% Penicilliin–Streptomycin and 1% L-glutamine. In addition, 15% FBS (Gibco, 10270106), 0.024 μg/mL LIF and 0.12 mM β-mercaptoethanol were added. Once MEFs were removed, cells were collected and cultured in untreated 10 cm² petri dishes at a density of 0.3 × 10⁶ cells per dish with EB medium containing IMDM (supplemented with 1% Penicillin–Streptomycin and 1% L-glutamine), 10% FBS (Gibco, 10270106), 0.6% Transferrin (Roche, 10652), 0.03% monothioglycerol (MTG) (Sigma, M6145) and 50 μg/mL ascorbic

acid (Sigma, A4544). After 3 days in culture, Flk1⁺ haemangioblast cells were isolated through magnetic activated cell sorting[24] using an anti-Flk1 APC conjugated antibody (Supplementary Table 3) and anti-APC microbeads (Miltenyi Biotec, 130-090-855).

For the haemangioblast differentiation, Flk1⁺ cells were cultured on gelatin for 24 to 72 h in IMDM medium supplemented with 1% Penicillin–Streptomycin, 1% L-glutamine, 15% FBS (Gibco, 10270106), transferrin (Roche), MTG (Sigma) and 50 mg/μL of ascorbic acid (Sigma), 15% D4T[24], 10 μg/mL VEGF (Preprotech, 500-P131) and 10 μg/mL IL6 (Preprotech, 216-16). The cells could then be collected with TrypLE express and cell populations analysed by flow cytometry using anti-VE-Cad, anti-CD41 and anti-Kit antibodies (Supplementary Table 3). For in vitro CD44–hyaluronan interaction experiments Flk1⁺ cells were plated on gelatin, as per the haemangioblast differentiation protocol, and incubated with either 10 μg/mL anti-CD44 antibody [KM201] (Supplementary Table 3), 300 μg/mL of hyaluronidase (Sigma, H4272) and 500 μM of 4MU (Sigma, M1381-25G). Generation of haematopoietic and endothelial populations was assessed by flow cytometry after 48 h in culture.

**Antibody screen, flow cytometry and cell sorting**. The antibody screen was performed using the Mouse Cell Surface Marker Screening Panel (BD Bioscience; Supplementary Table 3) according to the manufacturer's instructions. Cells from haemangioblast culture and dissociated AGMs were stained with different combination of antibodies (Supplementary Table 3). Dyes 7-AAD (Invitrogen, A1310) or Sytox Blue (Invitrogen, S34857) were used to exclude dead cells. FACS analysis was done using FACS Aria III (Becton Dickinson) and FACS Diva software. Data were later analysed using FlowJo v10.1r5 (Tree Star, Inc.). For single-cell qPCR analysis, cells were sorted using an 85 μm nozzle directly into the reaction buffer (Cells Direct qRT-PCR Kit, Invitrogen) and were snap frozen. For OP9 co-culturing assays, cells were sorted using 100 μm nozzle directly into a 96-well culture dish (Costar).

**Immunofluorescence and confocal microscopy**. Mid-gestation embryos were dissected and fixed in 4% paraformaldehyde for 15 min at room temperature then incubated in 15% sucrose solution for 1 h before freezing in OCT (Tissue-Tek). Ten-micrometer transverse cryo-sections of the AGM region were then placed on superfrost plus slides (Thermo Scientific). Sections were washed in PBS, incubated in 1 M glycine solution and permeabilized with 0.3% Triton X-100 (Sigma). Blocking solution consisting of 5% donkey serum, 5% goat serum and 0.1% Tween-20 (Sigma) in tris-buffered saline (TBS) buffer was applied to sections for 2 h at room temperature. Sections were incubated in primary antibodies overnight at 4 °C and then washed. Secondary antibodies (Thermo Scientific; Supplementary Table 3) were applied for 1 h at room temperature and were washed before 4′,6-diamidino-2-phenylindole nuclear stain (Invitrogen) was applied for 15 min. Slides were washed before being mounted with Prolong gold (Life Technologies) and imaged on a Leica SP5 confocal microscope.

**Single-cell q-RT-PCR**. Single cells were sorted directly into lysis buffer and snap frozen. Samples were reverse transcribed with superscript III reverse transcriptase from the Cells Direct one-step qRT-PCR Kit (Invitrogen) for 15 min at 50 °C. The cDNA was then pre-amplified for 20 cycles with 25 nM final concentration of each outer primer for a set of 96 target genes (Supplementary Table 2). The cDNA was then diluted with loading reagent (Fluidigm) and SoFastTM EvaGreen supermix (Biorad), and were loaded onto a chip with inner primer mix. Amplification of the target genes was measured with the Fluidigm Biomark HD system, with the Biomark Data Collection software and the GE96 × 96 + Meltv2.pcl programme.

**Single-cell qPCR data analysis**. Initial analysis of single-cell qPCR data were performed using the Fluidigm Real Time PCR analysis software[24]. Hierarchical clustering and PCA analysis were performed using the SINGuLAR analysis toolkit (Fluidigm version 3.5) in R software (version 3.2.1).

**Bulk RNA sequencing and data analysis**. Bulk RNA-seq was performed as described by the SmartSeq2 protocol[53]. Briefly, 25 cells from each group (Supplementary Table 4) were FACS sorted directly into lysis buffer containing 0.2% Triton X-100, oligo-dT primers and dNTP mix, and then snap frozen. Reverse transcription was then performed, followed by pre-amplification for 14 cycles. Nextera libraries were then prepared and sequenced on the Illumina Next Seq sequencer.

Sequencing data were analysed with the aid of the EMBL Galaxy tools (galaxy.embl.de)[54]—specifically FASTX for adaptor clipping, RNA STAR for mapping and htseq-count for obtaining raw gene expression counts. The R software (version 3.3.1, http://www.R-project.org) was then used to generate heatmaps and tSNE plots using the DESeq2, Scater, Biobase and pHeatmap packages.

**Cell cycle analysis**. To analyse the cell cycle status of AGM-derived cells, we used the Click-iT plus EDU flow cytometry kit (Life Technologies; C10633). After dissection, AGMs were incubated in 10 μM of EDU in PBS with 10% FBS for 1 h at 37 °C. The tissue was then washed and incubated in 1.25 mg/mL of collagenase for

30 min. The sample was washed again and stained for cell surface markers (CD44, VE-Cad and Kit; Supplementary Table 3). Approximately 1000 cells were sorted based on cell surface expression into their different populations. Cells were then washed in PBS with 1% bovine serum albumin and then fixed in paraformaldehyde. The cells were permeabilized with a saponin-based wash solution before the addition of a copper protectant and AF488 picolyl azide for the detection of EDU. Finally, the cell populations were stained with Hoechst 33342 (Invitrogen; R37605) before re-analysis on the FACS Aria III, to determine the proportion of cells from each population in G0/G1, S and G2 phases.

**OP9 co-culturing assays**. OP9 cells (ATCC, CRL-2749) were maintained in MEM-alpha medium (Gibco, 12561056) with 20% FBS (ATCC, 30-2020). For limiting dilution, co-cultures cells were sorted directly onto a confluent OP9 stromal layer and incubated in medium conducive to haematopoietic development, IMDM (Lonza) treated with 5% penicillin–streptomycin (Gibco) and supplemented with 10% FBS (Gibco, 10270106), L-glutamine, transferrin, MTG, ascorbic acid, LIF, 50 ng/mL SCF (Preprotech, 250-03), 25 ng/mL IL3 (Preprotech, 213-13), 5 ng/mL IL11 (Preprotech, 220-11), 10 ng/mL IL6 (Peprotech), 10 ng/mL Oncostatin M (R&D Systems, 495-MO-025), 1 ng/mL bFGF (R&D Systems, 233-FB-025). Round cell colonies were quantified after 3–6 days in culture. For ex vivo CD44 blocking antibody experiments, 20 VE-Cad+CD44$^{High}$ cells were sorted as per OP9 co-culturing assays, into a medium containing either no antibody, or 5 or 10 μg/mL of anti-CD44 antibody [KM201] (Supplementary Table 3). The number of colonies was assessed after 3–6 days in culture.

**Haematopoietic colony-forming assay**. One hundred cells were initially sorted onto a confluent OP9 stromal layer as per OP9 co-culturing assay. After 3 days in culture, cells were collected with TrypLE express (Gibco) and CFU-culture assays were initiated using Methocult complete medium (Stem Cell Technologies). Cells were grown in 35 mm culture dishes and colonies quantified after 7 days.

**Lymphocyte progenitor assay**. Fifty cells were sorted onto confluent OP9 (ATCC, CRL-2749) or OP9-DL1[55] stromal layers in MEM-alpha medium (Gibco) and supplemented with growth factors conducive to lymphocyte development, 20% FBS (ATCC, 30-2020), 50 ng/mL SCF (Peprotech), 5 ng/mL Flt-3L (Preprotech, 250-31L) and 1 ng/mL IL7 (Preprotech, 217-17). Medium was changed every 4–5 days and cells were split as necessary. Cells were cultured for 21 days before collecting, with TrypLE express for flow cytometry analysis.

**Reporting summary**. Further information on research design is available in the Nature Research Reporting Summary linked to this article.

## Data availability

The source data underlying Figs. 2a, e, 4b, 6c, 8c, f, 9b, c, f and Supplementary Fig. 17 are provided as a Source Data file.

All the expression data supporting the results reported in the article can be found in Supplementary Data 1–8. In addition, the sc-RNA-seq raw data are accessible from the ArrayExpress respository (E-MTAB-6987) and the bulk RNA-seq raw data were deposited at the NCBI Gene Expression Omnibus (GSE128971).

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

## Acknowledgements

We thank Kalina Stantcheva and Cora Chadick (EMBL Rome FACS Facility, Italy) for cell sorting; Andreas Buness (EMBL Rome Bioinformatics Services, Italy), Artem Adamov (EMBL, Italy) and Tallulah Andrews (Wellcome Trust Sanger Institute, UK) for help with bioinformatics analysis; Paul Collier (EMBL Genomics Core Facility, Germany) for bulk RNA-seq; Radvilė Prialgauskaitė (EMBL Rome) for help with immuno-fluorescence analysis; Gisela Luz (Patil Group, EMBL, Germany) for scientific illustra-tion; Inke Näthke (University of Dundee, Scotland), Paul Heppenstall (EMBL, Italy) and Alexander Aulehla (EMBL, Germany) for fruitful scientific discussions. The European Molecular Biology Laboratory and the Wellcome Trust supported this work.

## Author contributions

M.O.: conceptualization, formal analysis, investigation, and visualization, writing—original draft, writing—review and editing. Ö.V.B.: conceptualization, formal analysis, investigation, visualization, writing—review and editing. V.S.: formal analysis, visualization, writing—review and editing. M.S.: investigation, formal analysis, writing—review and editing. K.G.: investigation, visualization, writing—review and editing. K.Z.: formal analysis, visualization, writing—review and editing. P.V.P.: formal analysis, writing—review and editing. V.M.: formal analysis, writing—review and editing. V.F.: investigation, writing—review and editing. K.N.N.: investigation, writing—review and editing. B.B.: investigation, writing—review and editing. V.B., supervision, writing—review and editing. K.R.P.: supervision, investigation, writing— review and editing. S.A.T.: supervision, investigation, writing—review and editing. C.L.: conceptualization, formal analysis, supervision, investigation, visualization, methodology, writing—original draft, project administration, writing—review and editing.

## Competing interests

The authors declare no competing interests.
