## [Peer Review File · Nature Communications]

Reviewers' comments:

Reviewer #1 (Remarks to the Author):

Overall this is a very novel, detailed and interesting paper, that will advance the field. Overall it is well written and the conclusions substantiated. There are however a number of points that when addressed will improve the manuscript:

Whilst I agree that CD44 and CD61 split the population in two in Fig 1a, the rest of the antibodies are essentially either positive or negative. The text should be modified to reflect this.

Why is figure 1 a mixture of E11 and E10.5? This is confusing and the figure legend is also unclear when read with the text.

Whilst figure 2a is an impressive image, clearly showing two spatially distinct cell clusters expressing CD44, it is not able to show "various morphologies". A separate technique such as TEM would be required for this. Either additional information needs to be included, or the description of the data altered. The plots in 2b would be improved with the isotypes overlaid on the histograms. The gating should also be on real populations, not cutting through them (which I assume is based on an isotype control, but these should not be used in this way: they show specificity, but not absolute gating). It needs to be made clear in the legend if an independent experiment is an average of all the embryos in one pregnant mother and therefore the n = number of pregnant mothers assessed or something different. In e. it reads as if it is all from 1 litter. This needs to be across several separate litters. For d and e, the entire gating strategy needs to be shown, including a single cell and viability gate. Two tailed p test is an inappropriate statistical analysis for the data in e. If the data is normally distributed and has equal variance, then an ANOVA should be used.

The title on page 9 should be modified as the meaning of "an increasing haematopoietic profile" is very unclear.

The gates in the dotplot in figure 4a are not ideal as they are not around individual populations. Non rectangular gates and gates tightly around specific populations would be far more appropriate and would likely change the PC analysis.

The isotype control should be included in the blocking experiments in Figure 9a

Minor points:

Even if European legislation does not require ethics for the embryos, ethics would be needed for the pregnant mothers? This information should be included.

Reviewer #2 (Remarks to the Author):

This is an interesting manuscript that aims at characterizing how haemogenic endothelium (HE), the progenitor population of haematopoietic cells during developmental haematopoiesis, generates blood cells. The approach is rather multi-disciplinary since the authors use antibody screening and transcriptomic analysis (performed both in bulk and at single cells level) to guide their characterisation and involve work done with both mouse embryos and mouse embryonic stem cells (mESCs). The main findings are 1) that CD44 is a marker associated with the endothelial to haematopoietic transition (EHT) and its expression dynamics can define intermediate stages of this transition; 2) that the

CD44/hyaluronan axis is a regulator of EHT; and 3) that cells undergoing EHT are differentially metabolically regulated.

Whilst the question addressed is of potential interest, as a better characterization of this population is needed and could, in principle, be exploited for the in vitro generation of transplantable haematopoietic cells as well as for in vitro disease modelling, in my opinion there are some limitations that are left unanswered and need to be addressed to grant a publication in Nature Communications.

- The most important limitation regards the overall message. The authors use a very compelling title and in their introduction they continuously refer to haematopoietic stem cells (HSC). However, within the manuscript there is not a single transplantation assay performed to confirm that any of their findings are important for the generation of HSCs. All their conclusions are based on similarity of transcriptomic profiles, which is not enough to fully characterise populations that are highly heterogeneous, such as emerging blood cells in an embryo.

Of note, a good proportion of their mechanistic studies to validate their hypothesis of the importance of the CD44/hyaluronan axis in EHT is performed using mESCs differentiation, in particular using Flk1+ cells from day 3 EBs, which notoriously contain a mixture of primitive yolk sac (YS)-like progenitors as well as AGM-like cells.

Therefore, the absence of transplantation prevents the drawing of any conclusion about the importance of CD44 as a marker for EHT generating HSCs and of the downstream observation such as a differential metabolic profile of cells undergoing EHT and of hyaluronan as a critical regulator.

- Similarly, the comparison of the CD44 populations with pro-HSCs and pre-HSCs type I and type II is merely based on the clustering derived from in silico bioinformatic analysis. At the very least, an analysis of the expression of CD44 and Kit of the different pro- and pre-HSCs fractions should be shown to support their claims using multicolour flow cytometric analysis. It would be nonetheless preferable to analyse also the phenotypic progression of the lineage on sorted populations. In addition, since pro-HSC are already detected at E9.5, the authors should show a CD44/Kit plot instead of a CD44 histogram in Figure 2b.

- The analysis of the phenotypic progression would also help to dissect the precise stage requiring activation of the CD44/hyaluronan axis (Fig.9), which would strengthen even more the novelty of their finding.

- The final model shown in Fig 10 is not supported by the pattern of expression of CD44 in the dorsal aorta. In Fig.2A it is pretty clear that the entire dorsal aorta is CD44+, which includes a majority of endothelial cells. The collective body of their observations frankly support more a model where CD44 identifies endothelial cells lining the lumen of dorsal aorta, of which some undergo EHT. As such CD44 cannot be referred as a useful marker to identify HE, as hinted in the model, but rather CD44 show a dynamic range of expression during EHT.

- The metabolic findings are really quite interesting. Nonetheless the figure 6 is unintelligible, the authors should find a different way to present their data. In addition, some functional assay must be provided to support their claims. The authors should analyse the metabolic activity of cells determining for example basal oxygen consumption, glycolysis rates, ATP production and respiratory capacity of the different fractions.

Moreover, to support their claim of a non-proliferative quiescent state of the CD44^{low}Kit^{neg} cells, the author should analyse the cell cycle status of the different fractions in line with the recent report by Batsivari et al.

- This reviewer thinks that this is a well-written manuscript but that the figures can probably be condensed and that the parts describing the bioinformatic analysis with the extensive list of genes differentially expressed in the different populations affect the overall clarity of the work. These parts can probably be shortened.

Reviewer #3 (Remarks to the Author):

In this manuscript Oatley et al. investigated the endothelial to haematopoietic transition (EHT) during mouse development with particular focus on identifying the endothelial cells that give rise to haematopoietic cells. The study provided transcriptional profile of haemogenic endothelial cells and identified CD44 (together with Kit and VE-Cad) as a more reliable marker of different stages of EHT than the previously published method. In addition, authors show that CD44 plays a functionally important role in EHT.

The results section starts with the antibody screening which in my mind doesn't fit well with the flow of the study and doesn't add much scientifically. They next used scRNA-Seq to transcriptionally characterise VE-Cad⁺ cells, albeit on a rather small number of cells. This approach identified CD44 as a good marker for the haematopoietic cells co-expressing endothelial and haematopoietic genes.

It is not clear to this reviewer why authors switched to single cell qPCRs and bulk sequencing afterwards. Instead of sorting four different populations based on the CD44 and KIT expression they could have just sorted CD44⁺ and CD44⁻ and performed scRNA-Seq. This would be a more robust and unbiased way of assessing endothelial to haematopoietic transition and would allow the use of pseudotime ordering as a computational method to pinpoint the transition and relationship between different subpopulations. This is an important shortfall of the study. Having said that, the approach that authors used although not the best in this reviewer's opinion, it appears to be technically well executed.

There are just a few instances in which the conclusions are not matching the actual results and these should be corrected. Specifically:

1) I think authors should refrain from making comments related to the developmental relationship between different CD44 populations and just stick to describing the actual result which is defining transcriptional similarities between different clusters.

Page 10: "This suggests a developmental link between the CD44^{Low} populations where CD44^{Low}Kit^{Neg} cells would be the direct precursors of the CD44^{Low}Kit^{Pos} population which would then go on to generate CD44^{High} cells."

2) I am not sure that it is correct to conclude that all CD44⁺ cells have haematopoietic potential when no colonies were formed at the single-cell level from CD44^{Low}Kit^{Neg} population. It is only after sorting bulk of 300 cells that the colonies were observed. What is this saying about the frequency at which these cells can actually differentiate into blood?

Page 21: "Overall, we found that all populations expressing CD44 displayed haematopoietic differentiation capacity including CD44^{Low}Kit^{Neg} reinforcing the differentiation link between them as suggested by the transcriptome analyses described previously."

Overall, this is a technically well-performed study but the novelty is incremental rather than substantial.

Response to Reviewer #1 (Remarks to the Author):

Overall this is a very novel, detailed and interesting paper, that will advance the field. Overall it is well written and the conclusions substantiated. There are however a number of points that when addressed will improve the manuscript:

Whilst I agree that CD44 and CD61 split the population in two in Fig 1a, the rest of the antibodies are essentially either positive or negative. The text should be modified to reflect this.

CD44 and CD61 are indeed the most convincing markers because they split the VE-Cad+ population in negative and positive fractions of similar proportions. That said we respectfully disagree about the other markers. Even if there is a more disproportionate distribution of positive and negative cells, we find two fractions for each of the other six markers. CD93 and MADCAM1 are two proteins expressed by the majority of VE-Cad+ cells. However, we know based on our bulk RNA seq analysis that Cd93 and Madcam1 genes are expressed by endothelial cells and are down-regulated during EHT (Fig. 5a). Considering that cells with haematopoietic gene expression are scarce in the VE-Cad+ compartment, it is normal that the CD93 negative and MADCAM1 negative fractions are small compared to their positive counterparts. Another example is Sca1, which is a haematopoietic marker (see Ly6e, the corresponding coding gene, in Fig. 5a). We detected only 0.19% of VE-Cad+ Sca1+ cells while the VE-Cad+ Sca1- cells corresponded to 1.9% of the total cell population. Even if there is a large difference of frequency between these two subsets, Sca1 still split the VE-Cad+ population in two.

Why is figure 1 a mixture of E11 and E10.5? This is confusing and the figure legend is also unclear when read with the text.

EHT is known to take place in the AGM between approximately embryonic days 9.5 and 12. The choice of the different time points were done for practical reasons. We have chosen E11 for the validation of the antibody screen because more cells per AGM could be collected. At E10.5, there were less cells per AGM but we only needed 96 cells for the single-cell RNA sequencing experiment. In any case, with E10.5 and E11, we were in the right time window to capture the key stages of EHT. We have added in the legend the time point at which the RNA sequencing experiment was performed.

Whilst figure 2a is an impressive image, clearly showing two spatially distinct cell clusters expressing CD44, it is not able to show “various morphologies”. A separate technique such as TEM would be required for this. Either additional information needs to be included, or the description of the data altered.

Although we were able to show by FACS analysis that the different CD44⁺ populations were of different sizes, we agree with the Reviewer’s comment that not enough has been done to claim that CD44⁺ cells have “various morphologies.” We have altered the text to better reflect the results we obtained – “CD44 marks different cell populations in the AGM” and “Figure 2: CD44 splits the VE-Cadherin⁺ cells of the AGM into different populations.”

The plots in 2b would be improved with the isotypes overlaid on the histograms. The gating should also be on real populations, not cutting through them (which I assume is based on an isotype control, but these should not be used in this way: they show specificity, but not absolute gating).

Concerning the CD44 staining shown in 2b, given the number of fluorophores used in our panel, we opted to use fluorescence minus one (FMO) controls rather than the isotype to gauge background fluorescence in our samples. We have updated figure 2b to include the CD44 FMO control, which was used to differentiate between CD44⁻ and CD44⁺ cells in our experiments.

It needs to be made clear in the legend if an independent experiment is an average of all the embryos in one pregnant mother and therefore the n = number of pregnant mothers assessed or something different. In e. it reads as if it is all from 1 litter. This needs to be across several separate litters.

We agree with the Reviewer’s comment and have clarified the text for the figure 2 legend. The n values refer to independent litters of embryos, n = 4 indicates that litters from four pregnant mothers were assessed on different days.

For d and e, the entire gating strategy needs to be shown, including a single cell and viability gate.

We have provided the gating strategy for isolating CD44 populations in figure 2c and 2e. See Supplementary Figure 3a.

Two tailed p test is an inappropriate statistical analysis for the data in e. If the data is normally distributed and has equal variance, then an ANOVA should be used.

*We have re-analysed the data in Fig. 2e using instead an ANOVA and Tukey's HSD post-hoc test to determine significance. Figure 2 has been adjusted accordingly. A significant difference in cell size between the CD44 populations was found, ($F(3,16) = 142.01$, $p < 0.00001$). ** Represents a p-value of < 0.01 .*

ANOVA

Cell size

	SS	df	MS	F
Between Groups	9.3×10^9	3	3.1×10^9	142.01
Within Groups	3.4×10^8	16	2.2×10^7	
Total	9.6×10^9	19		

The title on page 9 should be modified as the meaning of “an increasing haematopoietic profile” is very unclear.

We have changed the title of this section. It now reads: “Detection of two CD44+ populations expressing blood genes”

The gates in the dotplot in figure 4a are not ideal as they are not around individual populations. Non rectangular gates and gates tightly around specific populations would be far more appropriate and would likely change the PC analysis.

Gating used in figure 4a was chosen based on our fluorescence minus one (FMO) controls and reflects the conditions we used to sort these rare haematopoietic populations. We could change the gates to reflect the clouds that appear in the FACS plot but this would not reflect what was sorted on the day. The sorting strategy was adopted to ensure we could capture the necessary number of cells for analysis and clouds are only visible after the sorting of the cells has taken place. From a sample of 0.835×10^6 cells we analysed, there were 168 Pro-HSCs, 244 Pre-HSCs type I and 708 Pre-HSCs type II observed. Given these low cell numbers it was not practical to use more stringent gating. We show in the figure below the index sorting data of the single cells (highlighted in blue) that were sorted for the single-cell qPCR analysis and subsequently used in the bioinformatic analysis.

The isotype control should be included in the blocking experiments in Figure 9a

To improve the robustness of our experiment in Figure 9 we added additional repeats using a different monoclonal anti-CD44 biotin antibody (clone IM7), whose epitope is positioned just outside of the link domain compared to the KM201 clone, which binds specifically to the link domain – the site of CD44-hyaluronan interaction. This second anti-CD44 antibody did not disrupt EHT. See Supplementary Figure 13.

Minor points:

Even if European legislation does not require ethics for the embryos, ethics would be needed for the pregnant mothers? This information should be included.

The welfare of adult mice used in this work was covered by the licence n° 17/2019-PR approved by the Italian Health Ministry. This information has been added to the method section.

Response to Reviewer #2 (Remarks to the Author):

This is an interesting manuscript that aims at characterizing how haemogenic endothelium (HE), the progenitor population of haematopoietic cells during developmental haematopoiesis, generates blood cells. The approach is rather multi-disciplinary since the authors use antibody screening and transcriptomic analysis (performed both in bulk and at single cells level) to guide their characterisation and involve work done with both mouse embryos and mouse embryonic stem cells (mESCs). The main findings are 1) that CD44 is a marker associated with the endothelial to haematopoietic transition (EHT) and its expression dynamics can define intermediate stages of this transition; 2) that the CD44/hyaluronan axis is a regulator of EHT; and 3) that cells undergoing EHT are differentially metabolically regulated.

Whilst the question addressed is of potential interest, as a better characterization of this population is needed and could, in principle, be exploited for the in vitro generation of transplantable haematopoietic cells as well as for in vitro disease modelling, in my opinion there are some limitations that are left unanswered and need to be addressed to grant a publication in Nature Communications.

- The most important limitation regards the overall message. The authors use a very compelling title and in their introduction they continuously refer to haematopoietic stem cells (HSC). However, within the manuscript there is not a single transplantation assay performed to confirm that any of their findings are important for the generation of HSCs. All their conclusions are based on similarity of transcriptomic profiles, which is not enough to fully characterise populations that are highly heterogenous, such as emerging blood cells in an embryo.

We thank the Reviewer for these valuable comments. We have not performed transplantations for the reasons listed below.

Previous studies have shown that:

1) *In the dorsal aorta, all Pre-HSCs and HSCs come from the VE-Cad⁺ fraction (Taoudi et al. 2008, PMID: 18593562; Rybtsov et al. 2011, PMID: 21624936).*

2) *Pro-HSC and Pre-HSC type I are able to reconstitute irradiated adult mice following a co-culture with OP9, while E11.5 Pre-HSC type II (VE-Cad⁺ CD45⁺) cells can directly reconstitute an irradiated adult mouse (Taoudi et al. 2008, PMID: 18593562; Rybtsov et al. 2011, PMID: 21624936 and Rybtsov et al. 2014, PMID: 25241746).*

3) *E11.5 AGM VE-Cad⁺ Gfi1⁺ cells can reconstitute irradiated adult mice both directly and following a co-culture with OP9 (Thambyrajah et al. 2016, PMID: 26619147).*

In this manuscript, we have shown the following:

1) *Pro-HSC, Pre-HSC type I and type II populations express the CD44 protein (Fig. 4b).*

2) *Clustering analysis performed using 96 genes carefully selected for their importance in EHT showed that Pro-HSC, Pre-HSC type I and type II expression profiles overlapped with our CD44⁺ Populations. A part of Pro-HSC (expressing blood genes) and Pre-HSC type I cluster with VE-Cad⁺ CD44^{Low} Kit⁺ while the Pre-HSPC type II group clusters with the VE-Cad⁺ CD44^{High} cells (Fig. 4c and 4b).*

3) *The VE-Cad⁺ CD44^{Low} Kit⁺ group is the only population expressing highly the Gfi1 gene (Fig. 3a and Supplementary Fig. 3).*

In conclusion, our populations overlap completely with very well-known cellular subsets, which have been shown to produce HSCs. This is why we considered that it was not necessary to perform transplantations to prove that the populations we have identified are relevant for HSC ontogeny.

Of note, a good proportion of their mechanistic studies to validate their hypothesis of the importance of the CD44/hyaluronan axis in EHT is performed using mESCs differentiation, in particular using Flk1⁺ cells from day 3 EBs, which notoriously contain a mixture of primitive yolk sac (YS)-like progenitors as well as AGM-like cells.

This is an important question. We would like to remind the Reviewer that the CD44KO mice do not have severe haematopoietic defects; compensatory mechanisms through other hyaluronan receptors may be at play to diminish the consequences of CD44 loss of function. This fact made it difficult to study the function of CD44 directly in vivo. To avoid compensatory effect and induce an acute disruption of CD44 binding to its ligand, we initially tested the effect of a CD44 blocking antibody on CD44^{High} differentiation in OP9 co-culture and noticed a clear negative effect on blood cell colonies formation (Fig. 9a, 9b,

9c). Unfortunately, this assay was not suitable for more in-depth analysis because of the very few cells that could be obtained. We decided to look for an alternative model to test the role of CD44 on EHT where we would have more cells to perform experiments. We eventually chose an in vitro ESC differentiation into blood model system for functional analysis of the CD44/hyaluronan axis. We did it for the following reasons:

1) A majority of endothelial cells produced in vitro were CD44 positive (Fig. 9d) like the AGM CD44^{Low} Kit^{Neg} population. Interestingly, the analysis of the mouse organogenesis sc-RNA-seq atlas (Cao et al. 2019 PMID: 30787437) showed that the *Cd44* gene is mostly expressed by a fraction of arterial endothelial cells between E9.5 and E13.5 indicating that this gene has a restricted expression pattern during development (Supplementary Figure 8). It makes its presence on endothelial cells in our in vitro model even more significant.

2) We have also shown in other studies that in vitro produced endothelial cells co-expressed the *Smad6* and *Smad7* genes as in the AGM CD44^{Low} Kit^{Neg} population (Bergiers et al. 2018 PMID: 29555020, Shvartsman et al. 2018 doi: <https://doi.org/10.1101/462978>, see heatmap below).

In this figure adapted from the supplementary figure 3 of Shvartsman et al. (2018), we show the expression by sc-q-RT_PCR of the indicated genes for three populations derived from ESC differentiation. SC_1 corresponds to endothelial cells co-expressing *Smad6* and *Smad7*, SC_2 and SC_3 correspond to cells co-expressing endothelial and blood genes. Most of SC_2 cells express *Gfi1* as well *Erg*, *Fli1*, *Lmo2*, *Lyl1*, *Fli1*, *Tal1*, *Runx1*.

3) The production of blood progenitors in vitro is also going through an intermediary stage expressing endothelial, haematopoietic genes, the key transcription factors Erg, Fli1, Lmo2, Lyl1, Fli1, Tall, Runx1 and Gfi1 as in the AGM (Bergiers et al. 2018 PMID: 29555020, see heatmap above).

4) Even though HSCs have not yet been generated from ESCs without the use of reprogramming, definitive erythroid, myeloid and lymphoid cells could be obtained from this model (Pearson et al. 2015, PMID: 25660408). These lineages could also be produced upon transplantation of in vitro derived progenitors in adult mice and could be detected up to 22 weeks after injection indicating a long-term reconstitution (Pearson et al. 2015, PMID: 25660408).

These elements convinced us that studying CD44 role in the in vitro haemangioblast culture would be relevant to AGM EHT.

The text of the manuscript has been modified to explain more clearly our reasons to use the in vitro ESC differentiation model into blood cells to study the role of CD44 in EHT.

Therefore, the absence of transplantation prevents the drawing of any conclusion about the importance of CD44 as a marker for EHT generating HSCs and of the downstream observation such as a differential metabolic profile of cells undergoing EHT and of hyaluronan as a critical regulator.

We have addressed the fact that we did not perform transplantations in pages 7 and 8.

- Similarly, the comparison of the CD44 populations with pro-HSCs and pre-HSCs type I and type II is merely based on the clustering derived from in silico bioinformatic analysis. At the very least, an analysis of the expression of CD44 and Kit of the different pro- and pre-HSCs fractions should be shown to support their claims using multicolour flow cytometric analysis.

As requested by the Reviewer, we analysed the co-expression of CD44 and Kit in Pro-HSC, Pre-HSC type I and type II (see figure below).

As expected, all these populations were Kit positive (Taoudi et al. 2008, PMID: 18593562; Rybtsov et al. 2011, PMID: 21624936 and Rybtsov et al. 2014, PMID: 25241746). In addition, they were all CD44 positive (Fig. 4b) as well, in accordance with our sc-q-RT-PCR data (Fig. 4c and Supplementary Fig. 5a).

It would be nonetheless preferable to analyse also the phenotypic progression of the lineage on sorted populations.

In Figure 7, we have data from *Runx1*^{-/-} AGM showing that both CD44^{Low}Kit^{Pos} and CD44^{High} populations were missing. Only the CD44^{Neg} and CD44^{Low}Kit^{Neg} populations remained in absence of *Runx1*. Considering the transcriptional similarities between CD44^{Low}Kit^{Neg} and CD44^{Low}Kit^{Pos} populations, we can hypothesise that the expression of *Runx1* in CD44^{Low}Kit^{Neg} cells would lead to the formation of CD44^{Low}Kit^{Pos} cells co-expressing endothelial and blood genes.

Our transcriptome and FACS analyses showed that CD44^{Low}Kit^{Pos} were similar to the Pre-HSC type I while the CD44^{High} were equivalent to Pre-HSC type II (VE-Cad^{Pos} CD45^{Pos}). Pre-HSC type I gives rise to type II in OP9 co-culture (Rybtsov et al. 2011, PMID: 21624936) and as expected, by isolating CD44^{Low}Kit^{Pos} cells, we could demonstrate that they were producing VE-Cad^{Pos} CD45^{Pos} on their way to produce blood cells supporting the differentiation relationship between the two CD44^{Pos}Kit^{Pos} populations (Supplementary Fig. 12).

In addition, since pro-HSC are already detected at E9.5, the authors should show a CD44/Kit plot instead of a CD44 histogram in Figure 2b.

We preferred to show a CD44 histogram because we did not have very many CD44+ cells following FACS analysis. Please see below the actual dot plots. The E10.5 time point has been included for comparison.

- The analysis of the phenotypic progression would also help to dissect the precise stage requiring activation of the CD44/hyaluronan axis (Fig.9), which would strengthen even more the novelty of their finding.

We agree that such an analysis would be an interesting addition to the study. However, as mentioned in our response on page 8, the co-culture assay was not suitable for more in-depth analysis of the CD44/hyaluronan axis because of the very few cells that could be obtained.

- The final model shown in Fig 10 is not supported by the pattern of expression of

CD44 in the dorsal aorta. In Fig.2A it is pretty clear that the entire dorsal aorta is CD44+, which includes a majority of endothelial cells.

Based on our microscopy image in Fig. 2a, it is not obvious to us that the entire aorta is CD44 positive. On the ventral side of the aorta, the CD44 staining is very convincing (left panel of the figure below). However, in the dorsal part of the vessel, we only saw weak scattered green dots (right panel of the figure below) that we interpreted as background staining.

The collective body of their observations frankly support more a model where CD44 identifies endothelial cells lining the lumen of dorsal aorta, of which some undergo EHT. As such CD44 cannot be referred as a useful marker to identify HE, as hinted in the model, but rather CD44 show a dynamic range of expression during EHT.

We still consider CD44 being a useful marker of endothelial cells undergoing EHT. For instance, the study by Zhou et al. (2016, PMID: 27225119) isolated VE-Cad⁺ cells from the AGM and found only endothelial cells lacking CD44 expression (see Supplementary Fig. 6). In retrospect, this is normal because CD44^{Neg} endothelial cells are far more abundant than CD44^{Low}Kit^{Neg} cells in AGM (see Fig. 2d). In addition, the recent atlas of mouse organogenesis has given us the possibility to examine CD44 expression on a wide range of embryonic endothelial cells between E9.5 and E13.5 (Cao et al. 2019 PMID: 30787437). Interestingly, CD44 was found mostly expressed by arterial endothelium (see Supplementary Figure 8 and figure below) but about 10 % of these cells (340 cells out of 3182) were expressing it. In our opinion, CD44 remains a useful marker to enrich for endothelial cells, which could undergo EHT.

- The metabolic findings are really quite interesting. Nonetheless the figure 6 is unintelligible, the authors should find a different way to present their data. In addition, some functional assay must be provided to support their claims. The authors should analyse the metabolic activity of cells determining for example basal oxygen consumption, glycolysis rates, ATP production and respiratory capacity of the different fractions.

Moreover, to support their claim of a non-proliferative quiescent state of the CD44^{low}Kit^{neg} cells, the author should analyse the cell cycle status of the different fractions in line with the recent report by Batsivari et al.

We have simplified the metabolism scheme (see Fig. 6). We have considered performing the battery of metabolite measurements that the Reviewer has suggested. Unfortunately, we were not able to collect sufficient amount of cells to perform this assay. This is a question that we will investigate in the future but we will need to have access to technologies sensitive enough to measure metabolites in very low number of cells.

A prediction from the metabolic transcriptional signature was that the CD44^{Low}Kit^{Neg} population would be resting. We therefore performed a cell cycle analysis of all four populations using EdU. Interestingly, our results matched very well with our expectations based on the metabolic transcriptional state that we found for our populations. The CD44^{Neg} population was significantly more cycling than the CD44^{Low}Kit^{Neg} subset of cells (see Figure 6c). Batsivari et al. (2017, PMID: 28479304) did not compare different AGM endothelial populations but they showed that there was an increase of cell proliferation along the progression into HSC. We observed also the same (see Figure 6c).

- This reviewer thinks that this is a well-written manuscript but that the figures can probably be condensed and that the parts describing the bioinformatic analysis with the extensive list of genes differentially expressed in the different populations affect the overall clarity of the work. These parts can probably be shortened.

We have simplified the Figure 3 and Figure 6.

Response to Reviewer #3 (Remarks to the Author):

In this manuscript Oatley et al. investigated the endothelial to haematopoietic transition (EHT) during mouse development with particular focus on identifying the endothelial cells that give rise to haematopoietic cells. The study provided transcriptional profile of haemogenic endothelial cells and identified CD44 (together with Kit and VE-Cad) as a more reliable marker of different stages of EHT than the previously published method. In addition, authors show that CD44 plays a functionally important role in EHT.

The results section starts with the antibody screening which in my mind doesn't fit well with the flow of the study and doesn't add much scientifically.

The purpose of the antibody screen was to find cell surface markers, which could be used to dissect the EHT process. The group of Prof. Cédric Blanpain has used this approach to successfully characterise the EMT process in cancer (Pastuskenko et al, 2018 PMID: 29670281). Although showing only single-cell RNA sequencing results would have made the manuscript simpler, we considered that presenting the results of two screening methods yielding overlapping marker genes was going to be much more convincing scientifically to the readers.

They next used scRNA-Seq to transcriptionally characterise VE-Cad⁺ cells, albeit on a rather small number of cells. This approach identified CD44 as a good marker for the haematopoietic cells co-expressing endothelial and haematopoietic genes.

It is not clear to this reviewer why authors switched to single cell qPCRs and bulk sequencing afterwards. Instead of sorting four different populations based on the CD44 and KIT expression they could have just sorted CD44⁺ and CD44⁻ and performed scRNA-Seq. This would be a more robust and unbiased way of assessing endothelial to haematopoietic transition and would allow the use of pseudotime ordering as a computational method to pinpoint the transition and relationship between different subpopulations. This is an important shortfall of the study. Having said that, the approach that authors used although not the best in this reviewer's opinion, it appears to be technically well executed.

This is an interesting point made by the Reviewer. Each technique we have employed was for answering a specific question. Single-cell transcriptome analysis with the CI platform allowed us to detect thousands of genes per cell in an unbiased manner to detect new cellular subsets and their corresponding marker genes. However, this method was low-throughput (only 96 cells) and noisy which made combining datasets computationally a very challenging task. This sc-RNA-seq work was performed at the end of 2014. At that time, there were no proper tools to combine computationally the results of multiple CI chips in a robust way.

To validate our initial results and characterise further our populations, we used single-cell q-RT-PCR. This method produces very robust and reliable data and has the power to detect low-abundant genes such as those coding for transcription factors. Although biased as we selected the genes to analyse, by employing this technique we could obtain accurate transcriptional data on the dynamics of many regulators known to be involved in EHT across our different populations. Furthermore, the data from single-cell q-RT-PCR could easily be combined across numerous experiments and there were no problem with batch effects or other technical issues associated with sc-RNA-Seq. This has enabled us to test the consistency of our sorting strategy and combine data from multiple experiments and time-points ensuring the reproducibility of our results.

To control for technical variation in our sc-qRT-PCR, we used a commercial cDNA mix allowing us to detect consistently most of our genes of interest (93 genes). In the bar graph below we have plotted the average Ct Values of the 93 genes from 19 independent PCR runs with the Fluidigm Biomark HD. The standard deviation is indicated as an error bar in the graph.

As shown in this plot, we had very little variation in performance across the different runs (average standard deviation 0.3).

Once we had characterised our populations in the context of EHT and found our sorting strategy to be reliable both in terms of gene expression, cell surface profile and culturing phenotype we sought to use a less biased approach in uncovering novel features of these cells. We opted for a 25 bulk RNA sequencing strategy to ensure we could stringently select the cells via FACS and to again enable us to detect low abundant genes expressed within our populations. This helped us to obtain good quality data with less noise than single cell

techniques and the depth of sequencing to understand the expression dynamics of low abundant genes.

Finally, an advantage of using multiple transcriptomics methods was that we could confirm our gene expression patterns with different technologies, which reinforced our confidence in our findings.

There are just a few instances in which the conclusions are not matching the actual results and these should be corrected. Specifically:

1) I think authors should refrain from making comments related to the developmental relationship between different CD44 populations and just stick to describing the actual result which is defining transcriptional similarities between different clusters.

Page 10: “This suggests a developmental link between the CD44Low populations where CD44LowKitNeg cells would be the direct precursors of the CD44LowKitPos population which would then go on to generate CD44High cells.”

In Figure 7, we have data from Runx1^{-/-} AGM showing that both CD44^{Low}Kit^{Pos} and CD44^{High} populations were missing. Only the CD44^{Neg} and CD44^{Low}Kit^{Neg} populations remained in absence of Runx1. Considering the transcriptional similarities between CD44^{Low}Kit^{Neg} and CD44^{Low}Kit^{Pos} populations, we can hypothesise that the expression of Runx1 in CD44^{Low}Kit^{Neg} cells would lead to the formation of CD44^{Low}Kit^{Pos} cells co-expressing endothelial and blood genes.

Our transcriptome and FACS analyses showed that CD44^{Low}Kit^{Pos} were similar to the Pre-HSC type I while the CD44^{High} were equivalent to Pre-HSC type II (VE-Cad^{Pos} CD45^{Pos}). Pre-HSC type I gives rise to type II in OP9 co-culture (Rybtsov et al. 2011, PMID: 21624936) and as expected, by isolating CD44^{Low}Kit^{Pos} cells, we could demonstrate that they were producing VE-Cad^{Pos} CD45^{Pos} on their way to produce blood cells supporting the differentiation relationship between the two CD44^{Pos} Kit^{Pos} populations (Supplementary Fig. 12).

2) I am not sure that it is correct to conclude that all CD44+ cells have haematopoietic potential when no colonies were formed at the single-cell level from CD44LowKitNeg population. It is only after sorting bulk of 300 cells that the colonies were observed. What is this saying about the frequency at which these cells can actually differentiate into blood?

Page 21: “Overall, we found that all populations expressing CD44 displayed haematopoietic differentiation capacity including CD44LowKitNeg reinforcing the

differentiation link between them as suggested by the transcriptome analyses described previously.”

We agree with the reviewer. We have no evidence that all CD44+ cells have haematopoietic potential. That is why we only referred to the CD44+ populations from which we could detect blood cell growth. The CD44^{Low}Kit^{Neg} population had a very low output indicating that cells with haematopoietic capacity are rare within it. We have edited the text so to make clear that this potential is very low.

Overall, this is a technically well-performed study but the novelty is incremental rather than substantial.

Reviewers' comments:

Reviewer #1 (Remarks to the Author):

The authors have addressed the majority of the issues raised and the manuscript is significantly improved. However, there are two issues which were raised originally that still required attention. 1). In regards to the response to the point: "Whilst I agree that CD44 and CD61 split the population in two in Fig 1a, the rest of the antibodies are essentially either positive or negative. The text should be modified to reflect

this." the response is not satisfactory and simply putting quads on a plot that go through populations and stating this divides them into positive and negative is not OK. If the authors are determined to split these into positive and negative, the proper isotype controls and overlapping histograms are essential and not just MFO's. This is essential for CD93, ICAM1, CD55.

2) The response to: "The gates in the dotplot in figure 4a are not ideal as they are not around individual populations. Non rectangular gates and gates tightly around specific populations would be far more appropriate and would likely change the PC analysis." is reasonable and understandable, but this information needs to be included in the manuscript and not just the response to the reviewers. This is important for the reader to understand why the gates are not optimal.

Reviewer #2 (Remarks to the Author):

I acknowledge that the authors made the effort to try to answer to all the questions that were raised during the first revision and overall the modification made to the manuscript have added clarity. However, despite this effort, this reviewer still disagrees with the interpretation of some of their data. In particular, in the absence of transplantation experiments, the authors cannot conclude that CD44 is a regulator of haematopoietic stem cell development as indicated in the title, which this reviewer still finds inaccurate.

In addition, there is quite a discrepancy in what the authors write in their response to the reviewers and what they write in the manuscript. In fact, the authors claim in page 13 of their rebuttal that the immunofluorescence shown in Fig.2A that CD44 show convincing staining only in the ventral part of the aorta and interpret the dotted pattern observed in the dorsal part as background.

Yet, in page 9 of the manuscript, they write "Overall, our results support the hypothesis that the CD44Neg cells and the CD44LowKitNeg cells belong to two distinct endothelial populations. The CD44Neg population expresses venous (Aplnr, Nr2f2 and Nrp2) and arterial (Sox17, Bmx and Efnb2) markers while the CD44LowKitNeg has a clear arterial signature with stronger expression of Bmx, Jag1 and Hey2".

Therefore, either CD44 dotted staining is real and the dorsal part of the aorta is composed by CD44LowKitNeg cells, or the dorsal part of the aorta comprises a majority of cells with venous identity.

This discrepancy is not only semantic, but it is important in light of the model proposed in Fig.10. This reviewer still believes that results shown do not support CD44 as a marker of haemogenic endothelium. Rather they show that CD44 expression is highly dynamic and increases during EHT.

Page 7. "Overall, we have demonstrated that the phenotypes based on CD44 expression could allow us to isolate all key populations in the process of HSCs formation more accurately than the phenotypes previously described"

Authors should change into "CD44 expression in combination with Kit"

Page 12. Interrupting hyaluronan binding to the CD44 disrupts EHT
Authors should change disrupts with "reduces".

Pag 13, the authors have added a lot of considerations that are more appropriate into discussion section.

Reviewer #3 (Remarks to the Author):

After reading the rebuttal letter I just have a minor comment. Authors should specify in the text what is the haematopoietic differentiation capacity of CD44 cells.

"Overall, we found that all populations expressing CD44 displayed haematopoietic differentiation capacity including CD44LowKitNeg albeit at a very low frequency."

Response to Reviewer #1 Comments:

The authors have addressed the majority of the issues raised and the manuscript is significantly improved. However, there are two issues which were raised originally that still required attention.

1). In regards to the response to the point: "Whilst I agree that CD44 and CD61 split the population in two in Fig 1a, the rest of the antibodies are essentially either positive or negative. The text should be modified to reflect

this." the response is not satisfactory and simply putting quads on a plot that go through populations and stating this divides them into positive and negative is not OK. If the authors are determined to split these into positive and negative, the proper isotype controls and overlapping histograms are essential and not just MFO's. This is essential for CD93, ICAM1, CD55.

After further consideration, we agree that the position of the quadrant for CD55 was cutting the cloud of positive cells. Regarding ICAM1, we repeated the experiment together with a specific isotype control and found very few ICAM1 negative cells (see below).

We therefore removed the FACS data for CD55 and ICAM1 antibodies. See Figure 1a and page 5.

Finally, CD93 was also repeated along with its specific isotype control and we still found two populations.

This was further supported by the expression pattern of Cd93 gene (Figure 5b).

2) The response to: "The gates in the dotplot in figure 4a are not ideal as they are not around individual populations. Non rectangular gates and gates tightly around specific populations would be far more appropriate and would likely change the PC analysis." is reasonable and understandable, but this information needs to be included in the manuscript and not just the response to the reviewers. This is important for the reader to understand why the gates are not optimal.

We have added the response to this specific comment in the text (pages 7 & 8).

Response to Reviewer #2 Comments:

I acknowledge that the authors made the effort to try to answer to all the questions that were raised during the first revision and overall the modification made to the manuscript have added clarity. However, despite this effort, this reviewer still disagrees with the interpretation of some of their data.

In particular, in the absence of transplantation experiments, the authors cannot conclude that CD44 is a regulator of haematopoietic stem cell development as indicated in the title, which this reviewer still finds inaccurate.

Indeed, we did not test specifically the impact of CD44 inhibition on HSC function in vivo. However, we showed that the CD44^{High} population (equivalent to the Pre-HSPC type II) demonstrated myeloid, erythroid and lymphoid potential based on our CFU-C and lymphoid assays (Figure 8). Moreover, we have shown in Figure 9a-c that that ex vivo treatment of CD44^{High} cells with the blocking anti-CD44 antibody diminished both the number of colonies formed on OP9 and their size.

Performing transplantations in this case would not have been helpful because CD44 is required for homing of HSCs (Cao H et al. 2016, PMID: 26546504). We do not know if the antibody treatment would have disrupted this process at the moment of the transplantation. Disentangling the role of CD44 in EHT and homing would be very challenging. It would be hard to conclude that a potential reduction of engraftment following anti-CD44 antibody treatment would be due to a differentiation problem and not to homing problems. Besides, performing transplantations would require more than six months of work in order to have a number of repetitions sufficient to reach statistical significance. In conclusion, we did not perform these experiments given that they are costly in terms of time and in number of animals and because the results would have been difficult to interpret.

When we used the term “regulator”, we used it in the sense that CD44 was regulating how efficiently the HSC development would occur. We did not mean that CD44 was a master regulator such as Runx1, a gene without which no HSCs can be generated (Okuda T et al. 1996, PMID: 8565077).

Nonetheless, in order to better reflect the content of our article, we propose to change the title to “Single-cell transcriptomics identifies CD44 as a new marker and regulator of endothelial to haematopoietic transition”. In that way, we encompass the full range of our results from in vivo/ex vivo analyses to the work we did with the in vitro embryonic stem cell differentiation system.

In addition, there is quite a discrepancy in what the authors write in their response to the reviewers and what they write in the manuscript. In fact, the authors claim in page 13 of their rebuttal that the immunofluorescence shown in Fig.2A that CD44 show convincing

staining only in the ventral part of the aorta and interpret the dotted pattern observed in the dorsal part as background.

Yet, in page 9 of the manuscript, they write “Overall, our results support the hypothesis that the CD44Neg cells and the CD44LowKitNeg cells belong to two distinct endothelial populations. The CD44Neg population expresses venous (*Aplnr*, *Nr2f2* and *Nrp2*) and arterial (*Sox17*, *Bmx* and *Efnb2*) markers while the CD44LowKitNeg has a clear arterial signature with stronger expression of *Bmx*, *Jag1* and *Hey2*”.

Therefore, either CD44 dotted staining is real and the dorsal part of the aorta is composed by CD44LowKitNeg cells, or the dorsal part of the aorta comprises a majority of cells with venous identity.

This discrepancy is not only semantic, but it is important in light of the model proposed in Fig.10. This reviewer still believes that results shown do not support CD44 as a marker of haemogenic endothelium. Rather they show that CD44 expression is highly dynamic and increases during EHT.

We thank the reviewer for raising this important point. We have added in the supplementary material an immunofluorescence analysis of CD44 expression in the dorsal aorta (see Supplementary Figure 2). We show data from E10 (same section as in Figure 2a) and from E11 (see below).

In both images, one can clearly see areas with lack of CD44 expression (see boxed areas). In contrast, the CD44+ cells of the aorta display a clear and defined fluorescence on the cell membrane. We still maintain that the arterial cells are not all positive for CD44. This is supported by single-cell RNA sequencing data (see Supplementary Figure 9).

In a previous publication titled “Molecular identification of venous progenitors in the dorsal aorta reveals an aortic origin for the cardinal vein in mammals” (Lindskog H et al. PMID: 24550118), it was shown that the dorsal aorta was heterogeneous during development. Indeed, it contained endothelial cells expressing venous markers at least until

E9.5. The authors did not analyse this heterogeneity further in development but this study supports the possibility that the dorsal aorta can be composed of endothelial cells of different nature at later time points.

We have assessed by immunofluorescence the expression of the venous marker APLNR in the dorsal aorta at E11 but did not detect APLNR+ cells (see below).

*It is possible that only the transcription of the *Aplnr* gene was initiated but that its protein was not produced. Studying a larger panel of venous specific proteins might lead to a different result. We have tried to detect NR2F2 but the antibody we have used did not work (not detected on venous endothelial cells), despite weeks of testing different conditions. We decided to stop, coming to the conclusion that it would require a significant time and money investment to find the antibodies able to work optimally in our experimental conditions.*

*Actually, our statement regarding the co-expression of arterial and venous markers by $CD44^{Neg}$ cells is in line with previous single cell RNA sequencing studies from the AGM (Zhou et al. PMID: 27225119 and Baron et al. PMID: 29955049). While analysing these datasets, we could identify $CD44^{Neg}$ cells. In the first analysis by Zhou, sixty-four per cent of $CD44^{Neg}$ cells (18 out of 28) were *Efnb2+Aplnr+* while fifty per cent of them (14 out of 28) were *Sox17+Aplnr+* (Supplementary Fig. 7). In the second dataset by Baron, thirty-six per cent of them (4 out of 11) were *Efnb2+Aplnr+* and eighteen per cent (2 out of 11) were positive for *Sox17* and *Aplnr* (Supplementary Fig. 8).*

Of note, not all of the $CD44^{Neg}$ cells co-expressed arterial and venous markers, suggesting heterogeneity within this population (Supplementary Fig. 7 & 8). This could be attributed to the fact that some of these endothelial cells come from the dorsal aorta (Supplementary

Fig. 2) and the rest from the cardinal veins, which are entirely $CD44^{Neg}$ (Supplementary Fig. 9). Indeed, cardinal veins are not removed when the AGM region is isolated because they are very close to the aorta (see figure below).

Image taken from Figure 6C of Prahst et al. (PMID: 24764455, 2014). Ao: Aorta, CV: Cardinal veins.

This would explain the presence of endothelial cells expressing only venous markers within the $CD44^{Neg}$ population.

We have changed the model and removed the termed “Hemogenic Endothelium” and replaced it by “ $CD44+$ Arterial Endothelium”. We kept the $CD44^{Neg}$ endothelial cells in the model but remove the arrow. Even if some of them come from the cardinal veins, they are from the AGM tissue that is a heterogeneous group of cells. Including the $CD44^{Neg}$ endothelial cells in the scheme is important because they provide an essential comparison point to reveal the key features (signalling, metabolic state, etc.) of $CD44+$ Arterial Endothelial cells from which HSPCs eventually emerge.

Page 7. “Overall, we have demonstrated that the phenotypes based on CD44 expression could allow us to isolate all key populations in the process of HSCs formation more accurately than the phenotypes previously described”

Authors should change into “CD44 expression in combination with Kit”
We modified the text as requested. See page 8.

Page 12. Interrupting hyaluronan binding to the CD44 disrupts EHT
Authors should change disrupts with “reduces”.
The change has been made (page 12).

Pag 13, the authors have added a lot of considerations that are more appropriate into discussion section.
The corresponding text has been moved to the discussion (page 16).

Response to Reviewer #3 Comments:

After reading the rebuttal letter I just have a minor comment. Authors should specify in the text what is the haematopoietic differentiation capacity of CD44 cells.

“Overall, we found that all populations expressing CD44 displayed haematopoietic differentiation capacity including CD44^{Low}Kit^{Neg} albeit at a very low frequency.”

We changed the text and wrote: “Overall, we found that all populations expressing CD44 displayed the capacity to produce haematopoietic cells on OP9 cells including CD44^{Low}Kit^{Neg} albeit at a very low frequency.” See page 12.

REVIEWERS' COMMENTS:

Reviewer #1 (Remarks to the Author):

The authors have addressed my remaining concerns

Reviewer #2 (Remarks to the Author):

After reading the rebuttal letter I am happy with the changes that the authors made.

Response to Reviewer #1 Comments:

The authors have addressed my remaining concerns

We are glad to have successfully addressed the concerns of the reviewer.

Response to Reviewer #2 Comments:

After reading the rebuttal letter I am happy with the changes that the authors made.

We are glad to have successfully addressed the concerns of the reviewer.